# Decoupled Entropy Minimization

**Jing Ma**[2*]**, Hanlin Li**[2*]**, Xiang Xiang**[1,2*]
[1] School of Computer Science and Tech, Huazhong University of Science and Tech, China
[2] School of AI and Automation, Huazhong University of Science and Technology, China

## Abstract

Entropy Minimization (EM) is beneficial to reducing class overlap, bridging domain gap, and restricting uncertainty for various tasks in machine learning, yet its potential is limited. To study the internal mechanism of EM, we reformulate and decouple the classical EM into two parts with opposite effects: cluster aggregation driving factor (CADF) rewards dominant classes and prompts a peaked output distribution, while gradient mitigation calibrator (GMC) penalizes high-confidence classes based on predicted probabilities. Furthermore, we reveal the limitations of classical EM caused by its coupled formulation: 1) reward collapse impedes the contribution of high-certainty samples in the learning process, and 2) easy-class bias induces misalignment between output distribution and label distribution. To address these issues, we propose **Ada**ptive **D**ecoupled **E**ntropy **M**inimization (AdaDEM), which normalizes the reward brought from CADF and employs a marginal entropy calibrator (MEC) to replace GMC. AdaDEM outperforms DEM*, an upper-bound variant of classical EM, and achieves superior performance across various imperfectly supervised learning tasks in noisy and dynamic environments.

## 1 Introduction

*Entropy Minimization* (EM) is a common self-supervised optimization method, which minimizes the conditional entropy of model predictions to reduce the class overlap [1]. It facilitates a low-density separation between classes and enhances confident predictions. EM has been shown to be useful for clustering, semi-supervised, and unsupervised learning [2; 3; 4]. Conditional entropy is also a widely validated calibration tool for Deep Neural Networks (DNNs), since it measures the prediction error and distribution shifts to some extent [5]. EM helps to bridge the domain gap and push the model's decision boundary towards the low-density regions of the target distribution. Therefore, EM is widely used in active learning, domain adaptation, and online learning as well [5; 6; 7; 8; 9].

Although Entropy Minimization is applied to a variety of tasks due to its simplicity and generality, previous literature indicates that the potential of classical EM is limited [10; 11; 12], causing unsatisfactory performance. To investigate the internal mechanisms by which EM effectively optimizes model parameters in an unsupervised manner, we attempt to reformulate and decouple the EM. We divide the conditional entropy in EM into two independent parts with entirely opposite effects: a positive-acting factor termed **C**luster **A**ggregation **D**riving **F**actor (CADF) and a negative-acting factor called **G**radient **M**itigation **C**alibrator (GMC). Minimizing CADF rewards dominant classes and promotes a peaked output distribution of the model, while minimizing GMC (log-sum-exp of the logits) penalizes the maximum logit and high-confidence classes based on predicted probabilities and calibrates the optimization direction, serving as a regularization term. This reformulation enables systematic analysis of these two parts and refining the classical EM, as shown in Fig. 1.

Importantly, we demonstrate that the highly coupled formulation of conditional entropy inherently constrains the effectiveness of classical EM. First, samples with lower confidence exert a more pro-

---

*Equal contribution, co-first author; Correspondence to `xex@hust.edu.cn` (also with Peng Cheng Lab)

nounced impact in the learning process, while the contribution of high-certainty samples diminishes significantly as their predicted probabilities approach 1.0, as shown in Fig. 2 (left). This phenomenon is named *reward collapse*, which is not preferable because samples with higher certainty can provide more reliable and informative signals for self-supervised learning. Second, classical EM tends to exhibit systematic bias toward dominant and easy classes, or assign most samples to a single cluster, resulting in severe misalignment between the model's output distribution and the ground-truth label distribution, as shown in Fig. 2 (right). *Easy-class bias* undermines the adaptability of classical EM in handling noisy or imbalanced tasks, particularly in dynamic or non-stationary environments.

Through the investigation in Sec. 3.2, we find that replacing the classical EM with minimizing CADF *alone* significantly enhances EM performance, yet it exhibits weaker robustness to target data distribution shifts. However, introducing a temperature $\tau$ to reshape the reward curve (by raising the value of $\tau$ when model's average predicted probability on target data increases) effectively improves CADF's robustness. Meanwhile, minimizing GMC helps reject erroneous highly confident predictions from poorly calibrated models [13]. Consequently, a weight $\alpha$ is employed to scale and control GMC's influence, mitigating model overfitting in noisy tasks and dynamic environments. **D**ecoupled **E**ntropy **M**inimization* (DEM*) inte-

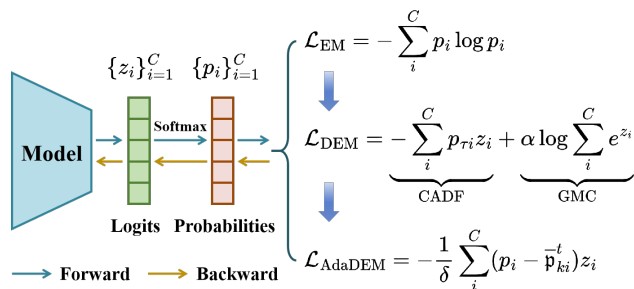

Figure 1: EM is decoupled into two parts with opposite effects: CADF and GMC. DEM* softens the model's prediction via temperature $\tau$ and scales GMC via weight $\alpha$, searching for optimal $(\tau*, \alpha*)$ to maximize classical EM's performance. AdaDEM normalizes CADF reward by $\delta$ (L1-norm of the gradients) to prevent reward collapse, and replaces GMC with Marginal Entropy Calibrator (MEC, *i.e.*, $\bar{\mathfrak{p}}_k^t$) to reduce easy-class bias.

grates CADF and GMC with searched optimal hyperparameters $(\tau^*, \alpha^*)$ in target data distributions. DEM* represents the performance *upper bound* of classical EM on the target task to some extent, but it incurs additional computational overhead for identifying optimal hyperparameters.

To address these issues, we propose **Ada**ptive **D**ecoupled **E**ntropy **M**inimization (AdaDEM). To enhance the relative contribution of high-certainty samples in the learning process, AdaDEM employs the L1-norm of rewards brought from CADF to normalize the conditional entropy. Additionally, a **M**arginal **E**ntropy **C**alibrator (MEC) is proposed to replace the GMC, which counteracts the overwhelming influence of dominant and easy classes by maximizing the estimated marginal entropy. Unlike existing methods that assume uniform label distributions [7; 12], MEC eliminates the need for label priors and instead leverages dynamic estimation during learning. In this way, AdaDEM, without requiring tunable hyperparameters, achieves comparable or even superior performance to DEM*.

In summary: 1) We provide an insightful view to study Entropy Minimization by reformulating and decoupling the classical EM into CADF and GMC with opposite effects. We investigate the roles of these two parts and reveal the inherent limitations of classical EM caused by its highly coupled formulation, *i.e.*, reward collapse and easy-class bias phenomena. 2) We propose DEM* to explore the performance upper bound of classical EM, and propose AdaDEM to mitigate reward collapse and easy-class bias while eliminating hyperparameter tuning requirements, thereby unleashing EM's potential. AdaDEM is validated to outperform DEM*. 3) Extensive experiments[2] across various tasks, including semi-supervised and unsupervised learning, domain adaptation, and reinforcement learning, demonstrate AdaDEM's superior performance. Additional evaluations on noisy/imbalanced benchmarks and dynamic/non-stationary environments further validate the effectiveness of AdaDEM.

## 2 Related Work

Shannon entropy [14] is a fundamental concept in information theory, which is used to measure the degree of uncertainty of a random variable. The higher the entropy value, the greater the uncertainty. For a discrete random variable $x$ with $n$ possible values $\{x_1, x_2, ..., x_n\}$ and corresponding prob-

---

[2]Source code is available at `https://github.com/HAIV-Lab/DEM`

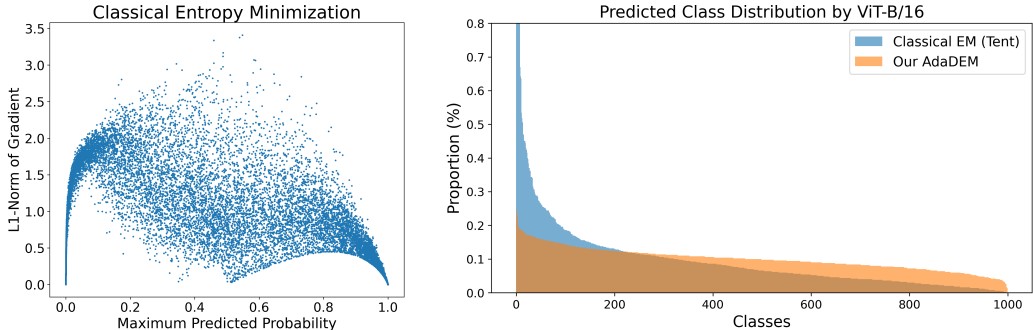

Figure 2: (**Left: Reward Collapse**) We compare the gradient magnitudes of the classical EM for samples with different predicted probabilities, which collapse to $0.0$ when the maximum probabilities approach $1.0$. (**Right: Easy-Class Bias**) The output distribution of ViT-B/16 after test-time adaptation using classical EM and our AdaDEM on a *class-balanced* Gaussian-noise-corrupted ImageNet-C benchmark, with all classes sorted by their predicted proportions in descending order.

abilities $\{p_1, p_2, ...p_n\}$, where $p_i \geq 0$ and $\sum_{i=1}^n p_i = 1$, the Shannon entropy $H(x)$ is defined as

$$H(x) = -\sum_{i=1}^n p_i(x_i) \log p_i(x_i). \tag{1}$$

In machine learning, Shannon entropy is generally used as an information-theoretic measure of uncertainty or impurity. It represents the amount of information needed to encode a distribution [15]. Given an input distribution $X$, a target distribution $Y$, and the sampled examples $\{x, y\}^N$. We build a model $f$ that maps $X$ to $Y$ that parameterized by $\theta$. The conditional output of $f$ is denoted as $p(y|x, \theta)$. Substituting it into Eq. (1), we can obtain the uncertainty of the model's output, which is defined as the expected value of the information carried by a sample from the output distribution:

$$H(Y|X) = -\frac{1}{N} \sum_x^X \sum_y^Y p(y|x, \theta) \log p(y|x, \theta). \tag{2}$$

The conditional entropy $H(Y|X)$ is a measure of class overlap [1]. Intuitively, if we want $p(y|x, \theta)$ to be highly peaked, *i.e.*, the model's prediction for $x$ is certain, we want $H(Y|X)$ to be low. On the other hand, if we want $p(y|x, \theta)$ to be flat or predictions to be uncertain, we can maximize the entropy $H(Y|X)$, which, in the limit, will lead to a uniform conditional distribution over classes [16].

By minimizing the conditional entropy $H(Y|X)$ for samples, the overlap of model's output distribution can be reduced, leading the density of data points to get lower at the decision boundary [17]. It facilitates a low-density separation between classes, a commonly prior assumption for semi-supervised learning [18]. A number of studies show that unlabeled data can be more informative if there is less overlap between classes [19]. Thus, EM has been demonstrated to be effective for clustering (avoiding trivial solutions where most instances concentrate in a single cluster) [2; 12; 20; 21], semi-supervised learning (enhancing model prediction accuracy per data point) [1; 3; 10; 22; 23; 24], and unsupervised learning (yielding peaked conditional class distributions) [4; 16; 20; 25; 26].

Deep Neural Networks (DNNs) suffer from distribution shifts. DNNs trained on the source domain tend to produce over-confident (low-entropy) predictions for in-distribution data and under-confident (high-entropy) predictions for out-of-distribution data [27]. Conditional entropy $H(Y|X)$ is a widely validated calibration tool for DNNs, since it measures the prediction error and distribution shifts to some extent. More confident predictions are all-in-all more correct. More severe shifts result in higher entropy [5]. One possible way to bridge the domain gap is to push the model's decision boundary towards the low-density regions of target distributions. On this basis, EM is widely used in active learning (reducing the number of possible hypotheses as rapidly as possible) [6; 15], domain adaptation (aligning the target data distribution with the source data distribution) [7; 27; 28], and online learning (connecting entropy to error and shift) [5; 8; 11; 29; 30].

Entropy Maximization plays a role in encouraging exploration in Reinforcement Learning [31; 32; 33]. Adding the entropy of policies to the objective function prevents premature convergence to suboptimal deterministic policies and is particularly beneficial for tasks that require hierarchical behavior

[31]. It is worth noting that maximizing the conditional entropy $H(Y|X)$ leads to a uniform output distribution to obtain uncertain predictions, while maximizing the marginal entropy, *i.e.*,

$$H(Y) = -\sum_{y}^{Y} p(y, \theta) \log p(y, \theta) \tag{3}$$

over all data points, where $p(y, \theta) = \frac{1}{N} \sum_{i=1}^{N} p(y|x_i, \theta)$, encourages the cluster sizes of classes to be uniform [12]. The combination of minimizing $H(Y|X)$ and maximizing $H(Y)$ makes the model's output distribution individually certain and globally diverse [7; 12; 16].

Unlike Cross-Entropy and KL-Divergence, which employ external supervision in the learning process, Entropy Minimization is a self-supervised learning paradigm. Therefore, addressing the potential limitations of classical EM can promote the development of imperfectly supervised machine learning. EM exhibits similar concepts to Self-Training [34] and Pseudo-Labeling [17], which leverage the model's own predictions for learning. Reward collapse and easy-class bias are also prevalent in these methods, and AdaDEM can be plug-and-play. While Early-Learning Regularization [35] mitigates label noise by using an exponential moving average of predictions as self-supervised signals, MEC in AdaDEM employs dynamic estimation of predictions to penalize dominant and easy classes.

## 3 Rethinking Entropy Minimization

### 3.1 Reformulating Conditional Entropy

**Notations.** For an example $x \in X$, the output of model $f(\theta)$ is $\mathbf{z} = [z_i]_{i=1}^{C} \in \mathbb{R}^C$. $z_i$ is the logit of the $i$-th class predicted by the model, and $C$ is the number of classes. We apply the Softmax function $\sigma(\cdot)$ to convert the real-valued vector $\mathbf{z}$ into a probability vector $\mathbf{p} = [p_i]_{i=1}^{C} \in \mathbb{R}^C$, *i.e.*, $p_i = e^{z_i} / \sum_{j=1}^{C} e^{z_j}$, where $0 \leq p_i \leq 1, \forall i \in \{1, 2, ..., C\}$.

For simplicity, we define the conditional entropy in Eq. (2) as

$$H(\mathbf{z}) = -\sum_{i=1}^{C} p_i(\mathbf{z}) \log p_i(\mathbf{z}). \tag{4}$$

The objective of EM is to minimize $H(\mathbf{z})$ as the sole loss function or a regularization term, namely,

$$\theta^* = \arg\min_{\theta} \sum_{x}^{X} H(\mathbf{z}|x, \theta) = \arg\min_{\theta} -\sum_{x}^{X} \sum_{i=1}^{C} p_i(\mathbf{z}|x, \theta) \log p_i(\mathbf{z}|x, \theta), \tag{5}$$

where $\theta^*$ is defined as the optimal solution for model $f(\theta)$.

To analyze the contributions of the two decoupled parts with opposite effects from classical EM to the model's output, we define the terms "reward" and "penalty." We define "reward" as the negative partial derivative of the loss function $\mathcal{L} = H(\mathbf{z})$ with respect to the logit $z_i$, *i.e.*, $-\partial\mathcal{L}/\partial z_i$, which aligns with the gradient descent direction for solving Eq. (5). We define "penalty" as the opposite of "reward", *i.e.*, $\partial\mathcal{L}/\partial z_i$. We subsequently demonstrate that minimizing CADF rewards the model's output logits, while minimizing GMC penalizes them.

**Reformulation.** To investigate the internal mechanisms by which EM effectively optimizes model parameters in an unsupervised manner, we attempt to reformulate and decouple the conditional entropy $H(\mathbf{z})$ in EM, which can be rewritten as

$$H(\mathbf{z}) = -\sum_{i=1}^{C} p_i(\mathbf{z}) \log \frac{e^{z_i}}{\sum_{j=1}^{C} e^{z_j}} = \underbrace{-\sum_{i=1}^{C} p_i(\mathbf{z}) z_i}_{\text{CADF}} + \underbrace{\log \sum_{i=1}^{C} e^{z_i}}_{\text{GMC}}. \tag{6}$$

As shown in Eq. (6), the conditional entropy is reformulated into two independent parts with entirely opposite effects. To analyze the impact of these two parts on EM, we derive the partial derivatives of the first term $T(\mathbf{z}) = -\sum_{i=1}^{C} p_i(\mathbf{z}) z_i$ and the second term $Q(\mathbf{z}) = \log \sum_{i=1}^{C} e^{z_i}$ with respect to $z_i$,

$$R_T = -\frac{\partial T(\mathbf{z})}{\partial z_i} = p_i(T(\mathbf{z}) + z_i + 1), \quad R_Q = -\frac{\partial Q(\mathbf{z})}{\partial z_i} = -\frac{e^{z_i}}{\sum_{j=1}^{C} e^{z_j}} = -p_i. \tag{7}$$

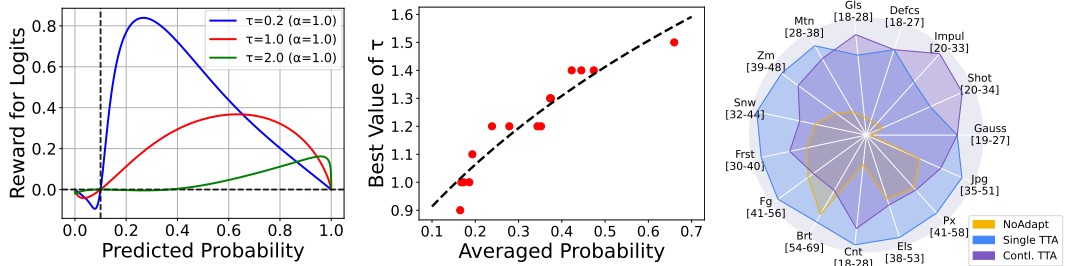

Figure 3: (**Left**) Reward curves of DEM with varying $\tau$ values for a 10-way classification task. (**Center**) The best $\tau$ value positively correlates with the average predicted probability of source models on target data. (**Right**) Detailed TTA results using the optimal $\tau$ across 15 target domains, with exact values sourced from Fig. 3 (Center). "NoAdapt" denotes the baseline using fixed source model parameters without adaptation, hence its performance remains consistent for both single-domain and continual TTA tasks. Values in brackets indicate the corresponding axis range.

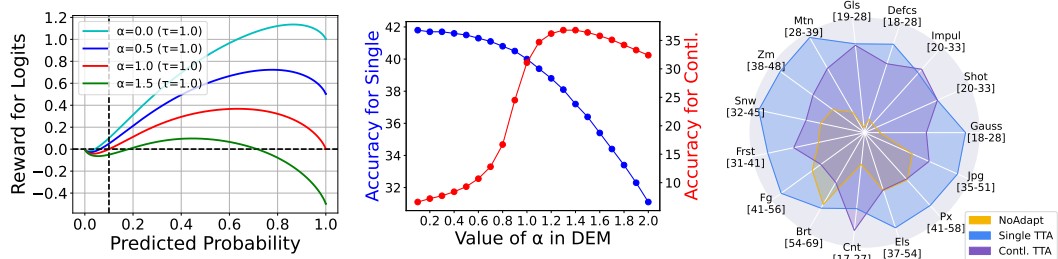

Figure 4: (**Left**) Reward curves of DEM with varying $\alpha$ values for a 10-way classification task. (**Center**) Effects of different $\alpha$ values on static and dynamic target data distribution shifts of single-domain and continual tasks. (**Right**) Detailed TTA results across 15 target domains, using the optimal $\alpha = 1.0$ for single-domain TTA tasks and $\alpha = 1.3$ for continual TTA tasks. These $\alpha$ values are selected based on the ablation results in Fig. 4 (Center). The definitions of "NoAdapt" and values in brackets are consistent with Fig. 3 (Right).

$R_T$ denotes the reward obtained by minimizing $T(\mathbf{z})$ for logits $z_i$, which leads to an positive increase in logit values. The reward curve of $R_T$ *w.r.t.* the probability $p_i$ is shown in Fig. 4 (left) ($\alpha = 0.0$ & $\tau = 1.0$). Minimizing $T(\mathbf{z})$ results in higher rewards for logits with elevated predicted probabilities, exhibiting an approximately linear trend. However, when reflected in probabilities through the Softmax function, this trend becomes amplified into exponential growth. Consequently, minimizing $T(\mathbf{z})$ causes the model to favor predicting classes with higher probabilities, thereby granting more rewards to dominant and easy classes. Conversely, $R_Q = -p_i \leq 0$ consistently imposes penalties on logits based on predicted probabilities. Minimizing $Q(\mathbf{z})$ causes higher-confidence classes to receive greater penalties, which drives the model's output distribution toward uniformity. So we name $T(\mathbf{z})$ the **C**luster **A**ggregation **D**riving **F**actor (CADF) and $Q(\mathbf{z})$ the **G**radient **M**itigation **C**alibrator (GMC). According to Eq. (6) and 7, we introduce $\sum_{i=1}^{C} \mathfrak{p}_i z_i$ that shares identical partial derivatives *w.r.t.* logits as $Q(\mathbf{z})$, where $\mathfrak{p}_i$ is a constant excluded from the computation graph[3].

Therefore, the conditional entropy can also be rewritten as

$$H(\mathbf{z}) = -\sum_{i=1}^{C} (p_i(\mathbf{z}) - \mathfrak{p}_i) z_i. \tag{8}$$

Next, we investigate the effects of CADF and GMC on entropy minimization and propose enhancements to address these limitations.

## 3.2 Decoupled Entropy Minimization

**Role of CADF and GMC.** To investigate the effects of CADF and GMC on EM, we conduct ablation studies on test-time adaptation (TTA) tasks using Tent [5]. Tent serves as an appropriate study subject

---

[3]In PyTorch, the "detach()" method can be used to obtain $\mathfrak{p}_i$

Table 1: (**Left**) Ablation studies in single-domain and continual TTA tasks. DEM* searches optimal hyperparameters $(\tau*, \alpha*)$ on a subset of target data. $\Delta$ denotes the performance improvement relative to the baseline. (**Right**) The sensitivity testing of learning rates demonstrates that AdaDEM has expanded 10x tolerance range compared to classical EM.

| Methods | Single | $\Delta$ | Contl. | $\Delta$ |
|---|---|---|---|---|
| *ResNet50* | | | | |
| NoAdapt | $31.5_{\pm0.00}$ | - | $31.5_{\pm0.00}$ | - |
| EM (Tent) | $40.0_{\pm0.03}$ | +0.0 | $31.2_{\pm0.11}$ | +0.0 |
| CADF | $41.7_{\pm0.05}$ | +1.7 | $36.1_{\pm0.03}$ | +4.9 |
| DEM* | $41.8_{\pm0.05}$ | +1.8 | $\mathbf{39.0}_{\pm0.02}$ | +7.8 |
| AdaDEM-Norm (w/o MEC) | $43.7_{\pm0.10}$ | +3.7 | $37.5_{\pm0.05}$ | +6.3 |
| AdaDEM-MEC (w/o Norm) | $\underline{44.4}_{\pm0.04}$ | +4.4 | $37.5_{\pm0.04}$ | +6.3 |
| AdaDEM (w/ MEC & Norm) | $\mathbf{44.8}_{\pm0.05}$ | +4.8 | $\underline{37.7}_{\pm0.05}$ | +6.5 |
| *ViT-B/16* | | | | |
| NoAdapt | $38.8_{\pm0.00}$ | - | $38.8_{\pm0.00}$ | - |
| EM (Tent) | $53.1_{\pm0.65}$ | +0.0 | $58.6_{\pm0.09}$ | +0.0 |
| CADF | $48.6_{\pm0.42}$ | -4.5 | $63.0_{\pm0.01}$ | +4.4 |
| DEM* | $56.0_{\pm0.32}$ | +2.9 | $\mathbf{64.1}_{\pm0.05}$ | +5.5 |
| AdaDEM-Norm (w/o MEC) | $54.0_{\pm0.45}$ | +0.9 | $59.3_{\pm0.04}$ | +0.7 |
| AdaDEM-MEC (w/o Norm) | $\underline{58.1}_{\pm0.23}$ | +5.0 | $62.4_{\pm0.14}$ | +3.8 |
| AdaDEM (w/ MEC & Norm) | $\mathbf{61.5}_{\pm0.20}$ | +8.4 | $\underline{63.2}_{\pm0.16}$ | +4.6 |

since it solely employs the classical EM as the loss function (where conditional entropy is measured from streaming data) to mitigate DNNs' performance degradation during online testing. Target data distribution shifts are categorized into *static* single-domain tasks and *dynamic* continual tasks. We adopt ResNet50 [36] and ViT-B/16 [37], two widely used DNNs pre-trained on ImageNet-1K [38], as source models for TTA starting points. Other implementation details are provided in Sec. 4.1. The experimental results in Table 1 demonstrate that replacing the classical EM with CADF *alone* significantly enhances EM performance, yet it suffers from poor robustness, as exemplified by the single-domain TTA case with ViT-B/16. To address these issues, we propose our improved solution.

**Reshaping the Reward Curve.** We introduce a temperature $\tau$ in CADF to soften the probability $p_i$, thereby reshaping the reward curve. This operation is defined in Eq. (9), with its partial derivative *w.r.t.* logit $z_i$ given in Eq. (17) (refer to Appendix A.2). Based on Eq. (17), the Fig. 3 (left) plots EM's reward curve across different $\tau$ values, which degenerates to classical EM when $\tau = 1.0$. Notably, the reward collapses to 0 when predicted probability approaches 1.0 or $1/C$. Introduction of $\tau$ ensures high-probability predictions remain within the high-reward interval, thus mitigating reward collapse.

$$T_\tau(\mathbf{z}) = -\sum_{i=1}^{C} p_{\tau i}(\mathbf{z})z_i, \qquad p_{\tau i}(\mathbf{z}) = \frac{e^{z_i/\tau}}{\sum_{j=1}^{C} e^{z_j/\tau}}. \tag{9}$$

We search the optimal $\tau$ to maximize performance on target tasks, and empirically demonstrate that the best $\tau$ value exhibits a positive correlation with the average predicted probability of source models on target data, $\frac{1}{N}\sum_{i=1}^{N} \max_{1\le j\le C} p_{ij}$, as shown in Fig. 3 (center). Detailed results in Fig. 3 (right).

**Scaling the Influence of GMC.** Note that $Q(\mathbf{z}) = \log\sum_{i=1}^{C} e^{z_i}$ takes the form of the log-sum-exp of the logits $z_i$, which provides a smooth approximation to the maximum of logits. Therefore, we introduce a weight $\alpha$ to control the penalties imposed by minimizing GMC on the maximum logit, preventing model overfitting caused by rapid growth in logit magnitudes, which is defined as

$$Q_\alpha(\mathbf{z}) = \alpha\log\sum_{i=1}^{C} e^{z_i}, \tag{10}$$

thereby scaling the reward of $Q_\alpha(\mathbf{z})$ to $R_{Q_\alpha} = -\alpha p_i$. We plot the reward curve of EM with varying $\alpha$ values, as shown in Fig. 4 (left). When $\alpha > 1.0$, EM with $Q_\alpha(\mathbf{z})$ penalizes logits corresponding to predicted probabilities near 1.0, helping to reject erroneous highly confident predictions from poorly calibrated models [13]. The results in Fig. 4 (center) demonstrate that slightly increasing the value of $\alpha$ can effectively mitigates model overfitting in noisy tasks and dynamic environments.

In summary, the **D**ecoupled **E**ntropy **M**inimization (DEM) is formulated as

$$H(\mathbf{z}) = T_\tau(\mathbf{z}) + Q_\alpha(\mathbf{z}) = -\sum_{i=1}^{C} p_{\tau i}(\mathbf{z})z_i + \alpha \log \sum_{i=1}^{C} e^{z_i}. \tag{11}$$

To explore the performance upper bound of classical EM, we search for the optimal combination within the search space defined by $\tau$ and $\alpha$ using a fast TPE algorithm [39]. In Proposition A.1, we theoretically prove that valid values of $\tau$ in DEM satisfy $0 < \tau \le 2/\alpha$ where $\alpha > 0$. DEM* searches the optimal hyperparameters $(\tau^*, \alpha^*)$ on a subset of target data. Implementations in Sec. 4.1.

### 3.3 Adaptive Decoupled Entropy Minimization

Classical EM suffers from reward collapse and easy-class bias phenomena, demonstrated in Fig. 2. DEM* enhances the contribution of high-certainty samples in the learning process by improving the CADF, while refining the GMC to penalize dominant and easy classes. However, DEM* cannot fundamentally resolve these limitations, as the dynamic interplay between model parameters updates and shifting data distributions causes the model's output to evolve over time. While continuous optimization or designing evolution strategies for hyperparameters $\tau$ and $\alpha$ remains feasible, such approaches incur substantial computational overhead, which is a drawback effectively addressed by the next proposed **Ada**ptive **D**ecoupled **E**ntropy **M**inimization (AdaDEM).

We employ the L1-norm of the reward brought from CADF, *i.e.*, $\delta = \| -\partial T(\mathbf{z}|x, \theta)/\partial \mathbf{z} \|_1$, to quantify the extent of changes in the model's output logits $\mathbf{z} = [z_i]_{i=1}^{C}$ before and after EM optimization, which is caused by updates to the model parameters $\theta$ induced by sample $x$. We adopt $1/\delta$ to normalize rewards across different deterministic samples, providing a more fundamental and direct approach compared to using the reciprocal of conditional entropy as the normalization factor [8; 11]. We have further verified that $\delta$ must be computed from the gradients of CADF, while using the gradients of the overall conditional entropy $H(\mathbf{z})$ would introduce penalties that compromise the accurate assessment of rewards brought by learning samples. Refer to Appendix D.6 for details.

Most existing methods for addressing easy-class bias employ a label prior to guide EM, typically by maximizing the marginal entropy to align the model's output distribution with a uniform label prior distribution [7; 12]. In contrast, we eliminate the label prior assumptions and propose a **M**arginal **E**ntropy **C**alibrator (MEC) to replace GMC. The MEC dynamically estimates the marginal entropy during learning via an exponential moving average of $1/N_k \sum^{N_k} \mathfrak{p}_k$ where $\mathfrak{p}_k \in \mathbb{R}^C$ is a probability vector of the $k$-th class. The estimated probability is dynamically updated as $\bar{\mathfrak{p}}_k^t = 0.9 \cdot \bar{\mathfrak{p}}_k^{t-1} + 0.1/N_k \cdot \sum^{N_k} \mathfrak{p}_k^t$ where $t$ denotes the iteration index and $\bar{\mathfrak{p}}_k^0 = [1/C]_{i=1}^{C} \in \mathbb{R}^C$.

In summary, the AdaDEM takes the form of

$$H(\mathbf{z}) = -\frac{1}{\delta} \sum_{i=1}^{C} (p_i(\mathbf{z}) - \bar{\mathfrak{p}}_{ki}^t)z_i, \quad k = \arg\max_i p_i(\mathbf{z}). \tag{12}$$

AdaDEM requires no tunable hyperparameters and achieves comparable or even superior performance to DEM*, as demonstrated in Table 1. Notably, AdaDEM also reduces the sensitivity of classical EM to learning rates, as shown in Table 1 (right). Detailed analyses are provided in Appendix D.

## 4 Experiments

### 4.1 Setups

**Benchmarks.** We conduct experiments on four benchmarks, *i.e.*, Test-Time Adaptation (TTA), Semi-Supervised Learning (SSL), Unsupervised Domain Adaptation (UDA), and Reinforcement Learning (RL) tasks. For TTA, we adopt ImageNet-C [40] containing 15 types of image corruptions, as well as ImageNet [38] and its variants: -A [41], -V2. [42], -R. [43], and -S. [44] that represent natural distribution shifts. For SSL, we consider CIFAR-10, CIFAR-100 [45], STL-10 [46], EuroSat [47], TissueMNIST [48], and Semi-Aves [49]. Regarding synthetic-to-real UDA, we mainly experiment with the semantic segmentation task, using GTA5 dataset [50] as the source domain and Cityscapes dataset [51] as the target domain. We employ Minigrid [52], a series of discrete environments, to conduct RL experiments. We also conduct experiments on class-imbalanced benchmarks of CIFAR-10-LT ($\rho = 100$) and CIFAR-100-LT ($\rho$=10) [45]. Refer to Appendix B.1 for detailed setups.

Table 2: Experiments on single-domain & continual TTA tasks (left) and the test-time prompt tuning task (right). Top-1 classification accuracy (%) is reported. We highlight the highest accuracy in **bold** and the second best as underline. Δ denotes the performance improvement relative to the baselines.

| Methods | Single-Domain Mean | Δ | Continual Mean | Δ |
|---|---|---|---|---|
| NoAdapt | 38.8±0.00 | - | 38.8±0.00 | - |
| Tent† | 53.1±0.65 | +0.0 | 58.6±0.09 | +0.0 |
| + DEM* | 56.0±0.32 | +2.9 | 64.1±0.05 | +5.5 |
| + AdaDEM | 61.5±0.20 | +8.4 | 63.2±0.16 | +4.6 |
| Tent | 52.7±0.10 | +0.0 | 48.5±0.71 | +0.0 |
| + DEM* | 55.1±0.11 | +2.4 | 64.5±0.14 | **+16.0** |
| + AdaDEM | 66.2±0.12 | **+13.5** | 64.4±0.02 | +15.9 |
| ETA | 65.1±0.10 | +0.0 | 64.2±0.04 | +0.0 |
| + DEM* | 66.3±0.04 | +1.2 | 65.7±0.04 | +1.5 |
| + AdaDEM | **66.8±0.02** | +1.7 | 66.1±0.01 | +1.9 |
| EATA | 62.2±0.14 | +0.0 | 64.9±0.08 | +0.0 |
| + DEM* | 64.4±0.30 | +2.2 | 66.2±0.07 | +1.3 |
| + AdaDEM | 65.3±0.11 | +3.1 | **66.4±0.04** | +1.5 |
| DeYO | 62.6±0.32 | +0.0 | 57.6±0.36 | +0.0 |
| + DEM* | 65.6±0.03 | +3.0 | 65.4±0.12 | +7.8 |
| + AdaDEM | 62.6±0.10 | +0.0 | 59.0±0.05 | +1.4 |
| SAR | 54.2±0.07 | +0.0 | 57.0±0.05 | +0.0 |
| + DEM* | 57.9±0.04 | +3.7 | 62.4±0.03 | +5.4 |
| + AdaDEM | 65.7±0.07 | +11.5 | 63.0±0.05 | +6.0 |

| Methods | ImageNet -1K | -A | -V2. | -R. | -S. | Avg. | Δ |
|---|---|---|---|---|---|---|---|
| *CLIP-RN50* | | | | | | | |
| Zero-Shot | 58.2±0.00 | 21.8±0.00 | 51.4±0.00 | 56.2±0.00 | 33.4±0.00 | 44.2±0.00 | +0.0 |
| Ensemble | 59.8±0.00 | 23.2±0.00 | 52.9±0.00 | **60.7±0.00** | 35.5±0.00 | 46.4±0.00 | +2.2 |
| TPT | 60.7±0.07 | 26.1±0.10 | 54.6±0.02 | 58.9±0.08 | 35.2±0.09 | 47.1±0.06 | +2.9 |
| + DEM* | 61.3±0.09 | 25.5±0.07 | 55.0±0.10 | 59.7±0.12 | 35.6±0.08 | 47.4±0.04 | +3.2 |
| + AdaDEM | 60.7±0.04 | 29.2±0.19 | 54.8±0.22 | 58.8±0.05 | 35.4±0.03 | 47.8±0.07 | +3.6 |
| CoOp | 63.3±0.00 | 23.1±0.00 | 55.4±0.00 | 56.6±0.00 | 34.7±0.00 | 46.6±0.00 | +2.4 |
| TPT (CoOp) | 65.4±0.06 | 28.9±0.14 | 58.2±0.10 | 59.0±0.09 | 36.3±0.15 | 49.6±0.07 | +5.4 |
| + AdaDEM | **65.6±0.05** | **31.3±0.10** | **58.5±0.22** | 59.3±0.10 | **36.3±0.11** | **50.2±0.06** | **+6.0** |
| *CLIP-ViT-B/16* | | | | | | | |
| Zero-Shot | 66.7±0.00 | 47.9±0.00 | 60.9±0.00 | 74.0±0.00 | 46.1±0.00 | 59.1±0.00 | +0.0 |
| Ensemble | 68.3±0.00 | 49.9±0.00 | 61.9±0.00 | 77.7±0.00 | 48.2±0.00 | 61.2±0.00 | +2.1 |
| TPT | 69.0±0.04 | 54.5±0.09 | 63.4±0.13 | 77.0±0.06 | 48.0±0.13 | 62.4±0.05 | +3.3 |
| + DEM* | 68.9±0.03 | 54.8±0.09 | 63.5±0.11 | 77.1±0.08 | 47.9±0.06 | 62.5±0.05 | +3.4 |
| + AdaDEM | 69.4±0.12 | 58.8±0.18 | 64.0±0.06 | 77.6±0.21 | 48.6±0.05 | 63.7±0.05 | +4.6 |
| CoOp | 71.5±0.00 | 49.7±0.00 | 64.2±0.00 | 75.2±0.00 | 48.0±0.00 | 61.7±0.00 | +2.6 |
| TPT (CoOp) | 73.6±0.05 | 57.9±0.12 | 66.9±0.08 | 77.2±0.04 | 49.2±0.07 | 64.9±0.06 | +5.8 |
| + AdaDEM | **73.7±0.07** | **60.3±0.11** | **66.9±0.19** | **77.9±0.14** | **49.3±0.07** | **65.6±0.02** | **+6.5** |

Table 3: Experiments on semi-supervised learning tasks. We report the average Top-1 classification accuracy (%) under different numbers of labeled samples. We highlight the highest accuracy in **bold** and the second best as underline. Δ denotes the performance improvement relative to the baselines.

| Methods | CIFAR-10 Mean | Δ | CIFAR-100 Mean | Δ | STL-10 Mean | Δ | EuroSat Mean | Δ | TissueMNIST Mean | Δ | Semi-Aves Mean | Δ |
|---|---|---|---|---|---|---|---|---|---|---|---|---|
| Ent. Min. | 96.4±1.9 | +0.0 | 72.6±0.5 | +0.0 | 82.4±1.6 | +0.0 | 76.5±3.6 | +0.0 | 47.3±2.6 | +0.0 | 59.9±0.7 | +0.0 |
| + AdaDEM | 97.2±0.2 | +0.8 | 75.8±0.3 | +3.2 | 84.8±0.3 | +2.4 | 83.7±0.8 | +7.2 | 49.3±1.3 | +2.0 | 61.0±0.2 | +1.1 |
| Vat (w/ Ent. Min.) | 97.4±2.0 | +0.0 | 77.5±0.7 | +0.0 | 85.4±0.9 | +0.0 | 86.6±5.4 | +0.0 | 45.2±4.8 | +0.0 | 61.0±0.4 | +0.0 |
| + AdaDEM | 98.5±0.1 | +1.1 | 78.8±0.7 | +1.3 | 86.9±0.1 | +1.5 | 91.2±0.9 | +4.6 | 49.4±0.6 | **+4.2** | 61.8±0.0 | +0.8 |
| MixMatch | 98.5±0.3 | +0.0 | 73.8±0.4 | +0.0 | 82.9±2.1 | +0.0 | 79.3±4.3 | +0.0 | 48.0±1.7 | +0.0 | 62.6±0.2 | +0.0 |
| + AdaDEM | 98.3±0.6 | -0.2 | 74.8±0.2 | +1.0 | 85.2±0.1 | +2.3 | 83.9±2.4 | +4.6 | **50.0±0.9** | +2.0 | 62.5±0.1 | -0.1 |
| FixMatch | 98.4±0.7 | +0.0 | 79.3±0.6 | +0.0 | 88.5±1.2 | +0.0 | 91.9±3.6 | +0.0 | 46.7±3.3 | +0.0 | **68.2±0.3** | +0.0 |
| + AdaDEM | 98.7±0.2 | +0.3 | 79.5±0.9 | +0.2 | 87.9±0.8 | -0.6 | **96.9±0.7** | +5.0 | 49.9±1.3 | +3.2 | 67.9±0.1 | -0.3 |
| FreeMatch | 98.9±0.0 | +0.0 | 82.6±0.1 | +0.0 | 89.7±1.6 | +0.0 | 95.8±1.0 | +0.0 | 45.6±3.3 | +0.0 | 67.0±0.3 | +0.0 |
| + AdaDEM | **99.0±0.1** | +0.1 | **83.1±0.1** | +0.5 | **91.6±0.2** | +1.9 | 95.9±0.1 | +0.1 | 47.9±0.4 | +2.3 | 67.3±0.2 | +0.3 |

**Implementations.** We consider the proposed DEM* and AdaDEM as alternatives to the classical EM. Therefore, we replace EM in the objective function of existing methods with our implementations of DEM* or AdaDEM. For instance, we substitute the loss function of Tent [5] with Eq. (11) or 12. DEM* employs a fast TPE algorithm [39] to search for the best hyper-parameters $(\tau^*, \alpha^*)$. Proposition A.1 helps to reduce the search scope. We use a subset comprising $\sim 20\%$ of test data with ground-truth labels for TPE, which is *only* applied for DEM* to explore the upper bound performance of classical EM. Note that AdaDEM requires neither test data labels nor hyperparameter tuning, strictly adhering to the unsupervised learning paradigm. We mainly utilize ResNet [36], ViT [37], CLIP Model [53], Deeplab-V2 [54], MLP and other DNNs as the research subjects. Meanwhile, for TTA, we adopt Tent [5], ETA [8], EATA [8], DeYO [11], SAR [29], TPT [30]; for SSL, we employ Ent. Min. [1], VAT [3], MixMatch [24], FixMatch [23], FreeMatch [22]; for UDA, we use MinEnt & AdvEnt [27]; and for RL, we take PPO [32] as baselines. We follow the implementations and hyperparameter setups of original methods unless otherwise specified. Uniformly, we report Top-1 accuracy for classification tasks, mIoU [55] for semantic segmentation tasks, and average return for RL tasks. All experiments are running in 3 seeds (*e.g.,* $\{1, 2, 3\}$) which control the orders of training/testing samples and the initial conditions of algorithms. Performance metrics are reported as "mean ± std". Refer to Appendix B.2 for detailed implementations.

Table 4: Experiments on unsupervised domain adaptation task of semantic segmentation. Standard mIoU (%) is reported.

| Methods | MinEnt | + AdaDEM | AdvEnt | + DEM* | + AdaDEM |
|---|---|---|---|---|---|
| mIoU | 41.6±0.47 | 42.7±0.13 | 43.6±0.19 | 44.6±0.32 | **44.9±0.17** |
| Δ | +0.0 | +1.1 | +0.0 | +1.0 | **+1.3** |

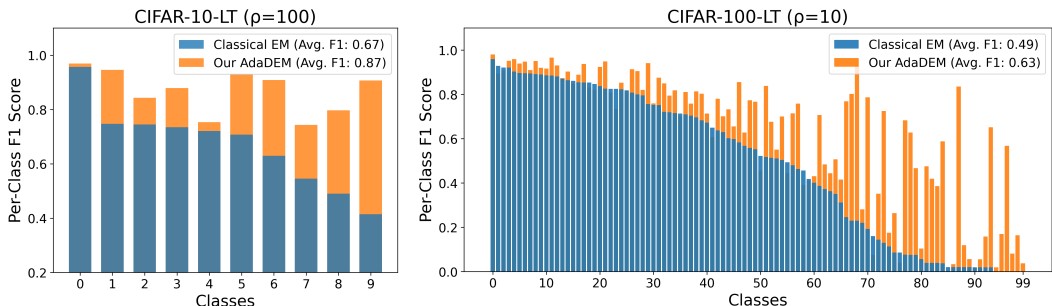

Figure 5: Experiments on class-imbalanced benchmarks. $\rho$ denotes the sample ratio between the most and least populous classes. Both per-class F1 scores and the average F1 score are reported.

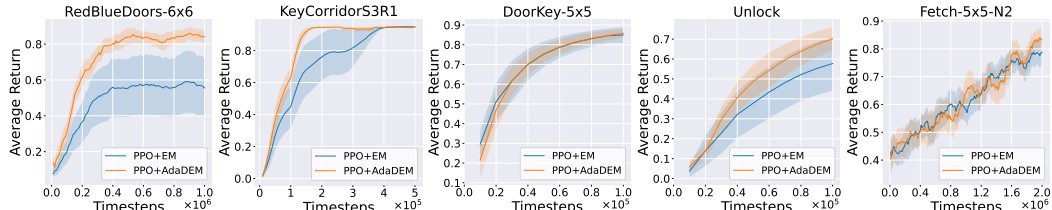

Figure 6: Experimental results on Reinforcement Learning tasks in Minigrid environments.

## 4.2 Main Results

In the following, we present the experimental results in various tasks. Details in Appendix C.

**Test-Time Adaptation.** As shown in Table 2, ETA, EATA, SAR, and DeYO are based on sample selection and assign larger weights to higher certainty test samples for mitigating the reward collapse. The application of DEM* and AdaDEM improves the original performance by addressing the potential limitations of the classical EM, which also enhances the prior-based EATA and SAR. For test-time prompt tuning task based on episodic TTA, DEM* and AdaDEM provide a solution suitable for the aggressive optimization strategy with large learning rate and multiple adaptation steps, which brings a significant improvement in unsupervised prediction to CLIP Models and TPT.

**Semi-Supervised Learning.** Entropy Minimization is widely used in SSL to facilitate low-density separation between classes. We set different numbers of available labeled samples, *i.e.*, $N_l$ per class, for the training of DNNs. Specifically, we set $N_l = 4/25/400$ for CIFAR-10, $N_l = 2/4/25$ for CIFAR-100, $N_l = 4/10$ for STL-10, $N_l = 2/4$ for EuroSat, $N_l = 10/50$ for TissueMNIST, and $N_l = 15 \sim 53$ for Semi-Aves. In Table 3, we report the average classification accuracy under different settings. We replace the classical EM in Ent. Min. and VAT (w/ Ent. Min.) with AdaDEM, and introduce AdaDEM as part of the loss functions in MixMatch, FixMatch, and FreeMatch, which do not use EM. We also validate on class-imbalanced benchmarks for SSL, as shown in Fig. 5, and provide an imbalance study on highly skewed data with varying degrees of imbalance severity, refer to Appendix D.10. Experimental results show that AdaDEM outperforms the classical EM. Meanwhile, incorporating AdaDEM into SSL methods effectively improves the performance of original algorithms.

**Unsupervised Domain Adaptation in Semantic Segmentation.** Beyond image classification, we also verify the effectiveness of DEM* and AdaDEM in the semantic segmentation task. As shown in Table 4, compared with the classical EM, AdaDEM can better bridge the domain gap between the source and target distributions, thus achieving a higher mIoU on the target data. For detailed results, refer to Table 10. DEM* and AdaDEM can improve the accuracy of tail classes and prevent the model from being inclined to predict dominant classes. We also visualize the segmentation results and pixel entropy in Fig. 7. AdaDEM reduces the prediction entropy of pixels more effectively, resulting in clearer object boundaries of out-of-distribution data.

**Reinforcement Learning.** Entropy *Maximization* is beneficial for encouraging agents' exploration in RL tasks. To demonstrate that our AdaDEM is not confined to the minimization strategy, we compare the effects of applying the classical EM and AdaDEM to PPO while keeping entropy maximization strategy unchanged. The experimental results, as depicted in Fig. 6, indicate that AdaDEM can achieve a higher or comparable average return.

### 4.3 Ablation Studies

We report the optimal hyperparameters $(\tau^*, \alpha^*)$ searched for DEM* in different methods and tasks in Table 11. For additional discussions, refer to Appendix D for details.

**Effect of Temperature $\tau$ in DEM*.** As shown in Fig. 3, increasing the value of $\tau$ promotes DNNs to learn high-certainty samples by reshaping the reward curve. The best $\tau$ for a target data distribution is positively correlated with the average prediction probability of DNNs for target samples.

**Effect of Weight $\alpha$ in DEM*.** As shown in Fig. 4 (left), GMC employs $\alpha$ to scale the penalty applied to logits, reducing the interference of over-confident samples for poorly calibrated DNNs. Decreasing the value of $\alpha$ is suitable for aggressive optimization strategies, while increasing $\alpha$ is suitable for alleviating the catastrophic forgetting caused by model overfitting, as shown in Fig. 4 (center).

**Sensitivity of AdaDEM to Optimizers and Learning Rates.** We compare the impacts of different optimizers and learning rates on classical EM and AdaDEM. As shown in Table 1, AdaDEM effectively reduces EM's sensitivity to the learning rate. Meanwhile, Table 2 demonstrates performance gains when applying AdaDEM to various optimizers including vanilla SGD (Tent$^\dagger$), Momentum (ETA, EATA, DeYO), Adam (Tent), and SAM (SAR). The results indicate that AdaDEM maintains compatibility with SGD or optimizers utilizing first-/second-order momentum estimation.

**Robustness of AdaDEM.** As shown in Table 2 and Fig. 5, source models' predictions on target data are significant noisy due to severe distribution shifts. Continual TTA tasks are evaluated under dynamically changing target data distributions. The results across these tasks demonstrate that AdaDEM exhibits robustness to noisy/imbalanced tasks and dynamic/non-stationary environments.

## 5 Conclusion

This paper provides an insightful view to study EM by reformulating and decoupling it into CADF and GMC with opposite effects. We reveal the limitations of classical EM caused by its highly coupled formulation: reward collapse and easy-class bias phenomena. We propose AdaDEM to overcome these limitations, which outperforms DEM*, an upper-bound variant of classical EM, and achieves superior performance across various machine learning tasks.

**Limitations.** DEM* and AdaDEM overcome the limitations of classical EM only from the perspective of objective functions. Although improving performance, they cannot completely solve all the problems of self-supervised learning. Additional targeted techniques and designs are required.

## Acknowledgements

This work was supported by the HUST Interdisciplinary Research Support Program (2025JCYJ077), the project of Peng Cheng Lab (PCL2025AS214), the 2026 Optics-Valley Excellence Project funded by the National Graduate College for Elite Engineers of HUST, the Natural Science Fund of Hubei Province (2022CFB823), and the School of Computer Science and Technology, School of Artificial Intelligence and Automation, Hoprcroft Center for Computing Science, and AI Institute, all at HUST.

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

# APPENDIX

## Contents

# A  More Design Details of DEM and AdaDEM

## A.1  Pseudo Code

---

**Algorithm 1** Pseudo code of DEM and AdaDEM in the PyTorch-like style

---

```python
# logits: the output logits of the model, in the shape of (N, C)
# tau: hyper-paramerter for CADF in DEM
# alpha: hyper-parameter for GMC in DEM
# avg_pred: the averaged predicted probabilities for C classes, initialized as None
# reset: a flag for resetting the MEC in AdaDEM, in bool type

import torch
import torch.nn.functional as F

# Decoupled Entropy Minimization (DEM)
p_tau = F.softmax(logits / tau, dim=1)

# CADF
cadf = - (p_tau * logits).sum(dim=1)

# GMC
gmc = alpha * torch.logsumexp(logits, dim=1)

# Total loss of DEM
dem_loss = cadf + gmc

# Adaptive Decoupled Entropy Minimization (AdaDEM)
p = F.softmax(logits, dim=1)
pseudo_label = p.argmax(dim=1)

# Initialize avg_pred to a (C, C) tensor of 1/C
if reset is True or avg_pred is None:
    C = torch.tensor(p.size(1), device=p.device)
    avg_pred = torch.ones((C, C), device=p.device) * 1.0 / C

with torch.no_grad():

    # Update avg_pred for MEC
    for label in torch.unique(pseudo_label):
        avg_pred[label] = 0.9 * avg_pred[label] +
                          0.1 * p[pseudo_label == label].mean(dim=0)

    # Calculate the L-1 norm of the rewards of CADF
    T = - (p * logits).sum(dim=1, keepdim=True)
    grad = (logits + T + 1) * p
    delta = grad.abs().sum(dim=1, keepdim=True)

# Original EM
# p = p - p.detach()
# AdaDEM-Norm
# p = (p - p.detach()) / delta
# AdaDEM-MEC
# p = p - avg_pred[pseudo_label]
# AdaDEM
p = (p - avg_pred[pseudo_label]) / delta

# Total loss of AdaDEM
adadem_loss = - (p * logits).sum(dim=1)
```

---

## A.2 Theory and Proof

Firstly, we present the detailed derivation process of reformulating and decoupling the conditional entropy for Eq. (6). Given the logit vector $\mathbf{z} = [z_1, ...z_i, ..., z_C] \in \mathbb{R}^C$ predicted by the model $f(\theta)$ for sample $x$ and the probability vector $\mathbf{p} = [p_1, ..., p_i, ..., p_C] \in \mathbb{R}^C$ obtained by applying the Softmax function $\sigma(\cdot)$. The conditional entropy can be rewritten as

$$
\begin{aligned}
H(\mathbf{z}) &= -\sum_{i=1}^{C} p_i \log p_i = -\sum_{i=1}^{C} p_i \log \frac{e^{z_i}}{\sum_{j=1}^{C} e^{z_j}} \\
&= -\left( \sum_{i=1}^{C} p_i z_i - \sum_{i=1}^{C} p_i \log \sum_{j=1}^{C} e^{z_j} \right) \\
&= -\left( \sum_{i=1}^{C} p_i z_i - \log \sum_{j=1}^{C} e^{z_j} \times \sum_{i=1}^{C} p_i \right) \\
&= -\left( \sum_{i=1}^{C} p_i z_i - \log \sum_{i=1}^{C} e^{z_i} \right) \\
&= \underbrace{-\sum_{i=1}^{C} p_i z_i}_{CADF} + \underbrace{\log \sum_{i=1}^{C} e^{z_i}}_{GMC},
\end{aligned}
\tag{13}
$$

where $p_i = e^{z_i} / \sum_{j=1}^{C} e^{z_j}$ and $\sum_{i=1}^{C} p_i = 1$. Note that $\log \sum_{j=1}^{C} e^{z_j}$ is independent of $i$.

Next, we derive the partial derivatives of the two independent parts in Decoupled Entropy Minimization (DEM) with respect to the logit $z_i$ for Eq. (7), respectively. According to the Softmax function, the partial derivatives of $p_i$ with respect to $z_i$ and $z_j$ are

$$
\frac{\partial p_i}{\partial z_i} = \frac{e^{z_i} \sum_{j=1}^{C} e^{z_j} - e^{2z_i}}{(\sum_{j=1}^{C} e^{z_j})^2} = p_i - p_i^2, \quad \text{and} \quad \frac{\partial p_i}{\partial z_j} = -\frac{e^{z_i} e^{z_j}}{(\sum_{j=1}^{C} e^{z_j})^2} = -p_i p_j. \tag{14}
$$

Let $T = -\sum_{i=1}^{C} p_i z_i$ denote the Cluster Aggregation Driving Factor (CADF), and let $Q = \log \sum_{i=1}^{C} e^{z_i}$ denote the Gradient Mitigation Calibrator (GMC). The partial derivatives of $T$ and $Q$ with respect to $z_i$ are calculated as follows,

$$
\begin{aligned}
\frac{\partial T}{\partial z_i} &= -\left( \frac{\partial z_i p_i}{z_i} + \sum_{j \neq i}^{C} \frac{\partial z_j p_j}{\partial z_i} \right) \\
&= -\left( p_i + z_i \frac{\partial p_i}{\partial z_i} + \sum_{j \neq i}^{C} z_j \frac{\partial p_j}{\partial z_i} \right) \\
&= -\left( p_i + z_i (p_i - p_i^2) + \sum_{j \neq i}^{C} z_j (-p_i p_j) \right) \\
&= -\left( p_i + p_i z_i - (p_i^2 z_i + \sum_{i \neq j}^{C} p_i p_j z_j) \right) \\
&= -\left( p_i + p_i z_i - \sum_{j=1}^{C} p_i p_j z_j \right) \\
&= -\left( p_i + p_i z_i - p_i \sum_{j=1}^{C} p_j z_j \right) \\
&= -p_i (T + z_i + 1),
\end{aligned}
\tag{15}
$$

$$\frac{\partial Q}{\partial z_i} = \frac{\partial log \sum_{j=1}^{C} e^{z_j}}{\partial z_i} = \frac{1}{\sum_{j=1}^{C} e^{z_j}} \frac{\partial \sum_{j=1}^{C} e^{z_j}}{\partial z_i} = \frac{e^{z_i}}{\sum_{j=1}^{C} e^{z_j}} = p_i. \tag{16}$$

Note that $T$ and $Q$ treat $z_i, \forall i \in \{1, 2, ..., C\}$ equally.

**Proposition A.1.** *The valid value of temperature $\tau$ in Decoupled Entropy Minimization is $0 < \tau \leq 2/\alpha$ where $\alpha > 0$.*

*Proof.* Considering the introduction of a temperature $\tau$ in CADF to reshape the reward curve, we obtain $T_\tau = -\sum_{i=1}^{C} p_{\tau i} z_i$, where $p_{\tau i} = e^{z_i/\tau} / \sum_{j=1}^{C} e^{z_j/\tau}$. Following the derivation process in Eq. (15), we calculate the partial derivative of $T_\tau$ with respect to $z_i$ as

$$\frac{\partial T_\tau}{\partial z_i} = -\frac{1}{\tau} p_{\tau i}(T_\tau + z_i + \tau). \tag{17}$$

Therefore, we can obtain the partial derivative of the conditional entropy used by DEM with respect to $z_i$ as

$$\frac{\partial H(\mathbf{z})}{\partial z_i} = \frac{\partial T_\tau}{\partial z_i} + \frac{\partial Q_\alpha}{\partial z_i} = \alpha p_i - \frac{1}{\tau} p_{\tau i}(T_\tau + z_i + \tau). \tag{18}$$

Let $F = T_\tau + z_i + \tau - \alpha \tau p_i / p_{\tau i}$, and we have $\partial H(\mathbf{z})/\partial z_i = -F \times p_{\tau i}/\tau$. We take the derivative of $F$ with respect to $z_i$ in order to derive the second derivative of $H(\mathbf{z})$ with respect to $z_i$:

$$\frac{\partial F_\tau}{\partial z_i} = 1 - p_{\tau i}(1 + \frac{z_i}{\tau} + \frac{T_\tau}{\tau}) - \alpha \tau (\frac{(\tau - 1) p_i}{\tau p_{\tau i}} - \frac{p_i^2}{p_{\tau i}} + \frac{p_i}{\tau}). \tag{19}$$

Under the boundary condition of $z_i = z_j = k, \forall\, i \neq j$, we have $p_i = p_{\tau i} = 1/C$, $\forall i \in \{1, 2, ..., C\}$, $T_\tau = -\sum_{i=1}^{C} p_{\tau i} z_i = -k \sum_{i=1}^{C} p_{\tau i} = -k$, and $F = \tau(1 - \alpha)$. Then, we can obtain the second derivative of $H(\mathbf{z})$ with respect to $z_i$ as

$$\begin{aligned}
\frac{\partial^2 H(\mathbf{z})}{\partial z_i} &= -\frac{p_{\tau i}}{\tau} \frac{\partial F}{\partial z_i} - \frac{F}{\tau} \frac{\partial p_{\tau i}}{\partial z_i} \\
&= -\frac{1}{\tau C}(1 - \frac{1}{C} - \tau \alpha(\frac{\tau - 1}{\tau} - \frac{1}{C} + \frac{1}{\tau C})) - (1 - \alpha) \times \frac{1}{\tau}(\frac{1}{C} - \frac{1}{C^2}) \\
&= (1 - \frac{1}{C})(\frac{\alpha}{C} - \frac{2}{\tau C}).
\end{aligned} \tag{20}$$

For the logits $\mathbf{z}$ predicted for a sample $x$, we assume that $z_m > z_j, \forall j \neq m$. It means that the probability of the $m$-th class is greater than others. We hope that the reward for $z_m$ is greater than zero, and the rewards for other $z_j, \forall j \neq m$ are less than zero, *i.e.*, $-\partial H(\mathbf{z})/\partial z_m > 0$ and $-\partial H(\mathbf{z})/\partial z_j < 0, \forall j \neq m$, in order to achieve low-density separation between classes and reduce the class overlap. Equivalently, we hope that $\partial^2 H(\mathbf{z})/\partial z_i \leq 0$ under the boundary condition of $z_i = z_j, \forall i \neq j$. Because $C > 1$ and $\alpha$ is set to be $\alpha > 0$, thus

$$\frac{\partial^2 H(\mathbf{z})}{\partial z_i} \leq 0 \iff \frac{\alpha}{C} - \frac{2}{\tau C} \leq 0 \iff \alpha \leq \frac{2}{\tau}. \tag{21}$$

When $\tau > 0$, we have $\tau \leq 2/\alpha$. The above inequality does not hold when $\tau < 0$.

Considering the limit conditions $\tau \to 0^+$ and $\tau \to 0^-$, we have

$$\lim_{\tau \to 0^+} \frac{\partial^2 H(\mathbf{z})}{\partial z_i} \to -\infty, \quad \text{and} \quad \lim_{\tau \to 0^-} \frac{\partial^2 H(\mathbf{z})}{\partial z_i} \to +\infty. \tag{22}$$

Overall, the valid value of $\tau$ in Decoupled Entropy Minimization is $0 < \tau \leq 2/\alpha$, where $\alpha > 0$.

$\square$

# B  More Implementation Details

## B.1  Experimental Protocols

### B.1.1  Single-Domain Test-Time Adaptation

ImageNet-C [40] is used to construct the single-domain TTA task. ImageNet-C contains 15 different versions of corruptions applied to $50,000$ images from the validation set of ImageNet-1K [38]. These corruptions are Gaussian noise (Gauss), Shot noise (Shot), Impulse noise (Impul), Defocus blur (Defcs), Frosted Glass blur (Gls), Motion blur (Mtn), Zoom blur (Zm), Snow (Snw), Frost (Frst), Fog (Fg), Brightness (Brt), Contrast (Cnt), Elastic (Els), Pixelation (Px), and JPEG (Jpg). Each type of corruption consists of 5 levels, with the 5-th level indicating the highest degree of damage. Following previous TTA methods [5; 8; 11; 29], we uniformly use the corruption at the 5-th level as the test datasets. We measure the Top-1 classification accuracy of TTA algorithms on each corruption subset and calculate the average accuracy as the metric for the single-domain TTA task. We employ 3 random seeds, to run experiments on ImageNet-C. These seeds determine the order in which test samples are seen by TTA algorithms and models over time. Since TTA is an online learning task, the arrival order of test samples can significantly impact the performance of TTA algorithms. We report the average accuracy and standard deviation across the 3 random seeds as metrics to measure the performance and robustness.

### B.1.2  Continual Test-Time Adaptation

Unlike the single-domain task tested separately on each corruption, continual TTA constructs a continuously changing testing environment using ImageNet-C [40]. Specifically, it concatenates the 15 corruptions in the 5-th level in the order of Gaussian noise $\rightarrow$ Shot noise $\rightarrow$ Impulse noise $\rightarrow$ Defocus blur $\rightarrow$ Frosted Glass blur $\rightarrow$ Motion blur $\rightarrow$ Zoom blur $\rightarrow$ Snow $\rightarrow$ Frost $\rightarrow$ Fog $\rightarrow$ Brightness $\rightarrow$ Contrast $\rightarrow$ Elastic $\rightarrow$ Pixelation $\rightarrow$ JPEG. This setup requires the TTA algorithms to have the ability to adapt aggressively in local static distribution shifts and maintain stability in the face of a dynamically changing environment without suffering from catastrophic forgetting. Similarly, we employ the Top-1 classification accuracy as the metric, conduct experiments with 3 random seeds, and report the average accuracy and standard deviation.

### B.1.3  Test-Time Prompt Tuning

We mainly use ImageNet-1K [38] and ImageNet variants, including ImageNet-A [41], ImageNet-V2 [42], ImageNet-R [43], and ImageNet-Sketch [44] as the test datasets. They are used to measure the robustness of TTA algorithms and models against natural distribution shifts. ImageNet-A consists of $7,500$ natural adversarial samples that are misclassified by the standard ResNet-50 [36] and contains 200 ImageNet categories. ImageNet-V2 includes $10,000$ images and $1,000$ ImageNet categories collected from the source different from ImageNet-1K. ImageNet-R collects $30,000$ artistic renditions from 200 ImageNet categories. ImageNet-Sketch is composed of $50,000$ black-and-white sketches, independently collected from the original ImageNet validation set, covering $1,000$ ImageNet categories. A main research object of test-time prompt tuning is the CLIP models [53], a series of foundational vision-language models. It uses test samples to update the input learnable prompts of the text encoder in CLIP models. Test-time prompt tuning focuses on performing episodic online learning on an individual test sample, that is, only one sample is used for prompt tuning at a time. The updated prompt is applied to make predictions on the current sample again, and then the learnable prompt is reset and waits for the next test sample to arrive. Similar to other TTA tasks, we employ the Top-1 classification accuracy as the metric and calculate the average accuracy and standard deviation over 3 random seeds.

### B.1.4  Semi-Supervised Learning

We use CIFAR-10 [45], CIFAR-100 [45], STL-10 [46], EuroSat [47], TissueMNIST [48], and Semi-Aves [49] as the datasets for evaluating SSL algorithms. CIFAR-10 and CIFAR-100 contain $50,000$ $32 \times 32$ training images and $10,000$ test images, covering 10 and 100 categories respectively. STL-10 consists of $13,000$ $96 \times 96$ RGB images, with $5,000$ labeled images and $100,000$ unlabeled images for training, and $8,000$ for testing. The EuroSAT dataset focuses on land use and cover classification, which includes 10 categories, $16,200$ training images, and $5,400$ test images. The

TissueMNIST dataset is related to biomedical images, mainly containing 8 biomedical scenarios and tissue types, with $165,466$ $28 \times 28$ training images and $47,280$ test images. The Semi-Aves dataset is a bird dataset for semi-supervised image classification, extracted from the iNaturalist-2018 dataset, containing 200 bird species, $3,959$ labeled images and $26,640$ unlabeled images for training, and $4,000$ for testing, with an average of $15 \sim 53$ labeled images per class. We strictly follow the configuration of various comparison methods and datasets in USB[4] [56], a unified semi-supervised learning benchmark. We report the average Top-1 classification accuracy and standard deviation from running experiments on 3 random seeds.

### B.1.5 Unsupervised Domain Adaptation in Semantic Segmentation

We focus on the synthetic-to-real unsupervised domain adaptation task, in which the model is trained on fully annotated synthetic images and validated on real-world data. Several unlabeled real images are accessible during training. We use the GTA5 dataset [50] as the source domain data, which contains $24,966$ synthetic frames and pixel-level semantic annotations of 33 categories captured from a video game. The Cityscapes dataset [51] is used as the target domain data, and its $2,975$ unlabeled images are utilized for training. The 19 common categories of GTA5 and Cityscapes are selected. We measure the segmentation performance with the standard mean-Intersection-over-Union (mIoU) metric [55], evaluated on $500$ validation images. We also report the average mIoU and standard deviation over 3 random seeds.

### B.1.6 Reinforcement Learning

We mainly consider RL tasks in discrete environments and utilize the Minigrid environment [52]. Minigrid offers a series of grid environments with different layouts and tasks, enabling agents to move within these grids. The agents' actions are typically discrete, such as basic actions like moving up, down, left, and right. We conduct experiments in 9 environments, including DoorKey-5x5, Empty-Random-5x5, Fetch-5x5-N2, FourRooms, GoToDoor-5x5, KeyCorridorS3R1, PutNear-6x6-N2, RedBlueDoors-6x6, and Unlock. We employ the RL Baselines3 Zoo[5], a training framework for Stable Baselines3 RL agents, to run our experiments and strictly follow the configurations of RL algorithms implemented therein. We report the mean and standard deviation of the average return over 10 episodes, which is measured on 6 random seeds.

## B.2 Method Implementations

We primarily run experiments on one NVIDIA GeForce RTX 4090 GPU with 24 GB of memory.

**DEM\*.** In Sec. 3.2, we propose DEM and introduce two hyperparameters, $\tau$ and $\alpha$, to improve the classical EM. When $\tau = 1.0$ and $\alpha = 1.0$, DEM is completely equivalent to classical EM. To find the most suitable hyperparameter configuration, we construct a search space where $\tau$ and $\alpha$ range from $0.0$ to $2.0$ (including $0.0$ and $2.0$), sampled at intervals of $0.1$. Through TPE [39], a fast hyperparameter search algorithm integrated in NNI[6], we can search for the optimal hyperparameter configuration $(\tau^*, \alpha^*)$. We call this method DEM\*. However, when applying DEM\* to replace or add to the loss functions of existing TTA and SSL algorithms, it requires a significant amount of additional computational overhead to search the suitable hyperparameters. Although we do not recommend this, if necessary, the way of introducing and searching for $(\tau^*, \alpha^*)$ in DEM\* can also be applied to AdaDEM, which is discussed in Appendix D. We share our practical experience in tuning $\tau$ and $\alpha$ for quick hyperparameter searching. For DEM, changes in the value of $\alpha$ have a greater impact on the original algorithm than changes in the value of $\tau$. This means that we can fix $\tau = 1.0$ and search for the optimal $\alpha^*$, then fix $\alpha^*$ and search for the optimal $\tau^*$, thus obtaining a sub-optimal hyperparameter combination. We also notice that the valid value of $\tau$ in DEM is $0 < \tau \le 2/\alpha$, which is considered to help determine the search scope of $\tau$.

**AdaDEM.** To avoid the complications brought by hyperparameter search, we do not introduce $\tau$ and $\alpha$ in AdaDEM, even though it is feasible, as discussed in Appendix D. We uniformly use the loss function in Eq. (12) to replace or add to existing TTA and SSL algorithms. Specifically, if the original algorithms employ the Entropy Minimization loss as a part of the objective function, we

---

[4]`https://github.com/microsoft/Semi-supervised-learning`
[5]`https://github.com/DLR-RM/rl-baselines3-zoo`
[6]`https://github.com/microsoft/nni`

Table 5: Detailed experimental results on single-domain TTA task. Top-1 accuracy (%) is reported. We highlight the highest accuracy in **bold** and the second best as underline. $\Delta$ denotes the performance improvement relative to the baselines.

| Methods | Single-Domain TTA | | | | | | | | | | | | | | | Mean | $\Delta$ |
| | | Noise | | | Blur | | | | Weather | | | | Digital | | | | | |
| | Gauss | Shot | Impul | Defcs | Gls | Mtn | Zm | Snw | Frst | Fg | Brt | Cnt | Els | Px | Jpg | | |
|---|---|---|---|---|---|---|---|---|---|---|---|---|---|---|---|---|---|
| NoAdapt | 29.1 | 29.6 | 31.6 | 31.2 | 25.1 | 39.3 | 31.5 | 24.6 | 30.2 | 54.3 | 64.5 | 48.4 | 34.2 | 52.5 | 55.1 | 38.8±0.00 | - |
| Tent[†] | 54.2 | 54.8 | 55.4 | 56.6 | 54.0 | 60.8 | 34.2 | 3.8 | 9.6 | 70.9 | 76.5 | 69.4 | 59.5 | 70.0 | 67.2 | 53.1±0.65 | +0.0 |
| + DEM* | 51.7 | 51.9 | 53.3 | 54.8 | 51.2 | 58.7 | 50.8 | 14.7 | 44.3 | 69.9 | 77.0 | 68.5 | 57.3 | 69.0 | 66.8 | 56.0±0.32 | +2.9 |
| + AdaDEM | 57.2 | 58.2 | 58.1 | 58.9 | 60.2 | 65.5 | 63.3 | 2.0 | 66.0 | 73.7 | 78.0 | 68.7 | 69.2 | 73.3 | 70.4 | 61.5±0.20 | +8.4 |
| Tent | 51.6 | 52.0 | 53.1 | 52.3 | 47.6 | 56.6 | 46.7 | 10.2 | 29.7 | 67.2 | 74.2 | 67.2 | 50.9 | 66.5 | 64.3 | 52.7±0.10 | +0.0 |
| + DEM* | 50.1 | 50.1 | 51.7 | 52.1 | 46.6 | 55.6 | 46.3 | 30.0 | 52.2 | 67.5 | 75.8 | 67.0 | 50.4 | 66.1 | 64.2 | 55.1±0.11 | +2.4 |
| + AdaDEM | **57.5** | 58.5 | 58.6 | 58.3 | 60.3 | 66.3 | 65.3 | 63.8 | 67.8 | 73.6 | 78.5 | 67.8 | 70.8 | **74.2** | **71.7** | 66.2±0.12 | **+13.5** |
| ETA | 56.2 | 57.1 | 57.3 | 58.3 | 58.8 | 63.9 | 61.2 | 66.6 | 65.9 | 73.4 | 77.6 | 70.0 | 67.2 | 72.4 | 70.1 | 65.1±0.10 | +0.0 |
| + DEM* | 57.3 | 58.2 | 58.3 | **59.7** | 60.0 | 65.4 | 62.8 | 68.2 | 67.0 | 74.2 | 79.1 | 70.7 | 69.0 | 73.9 | 71.4 | 66.3±0.04 | +1.2 |
| + AdaDEM | 57.4 | **58.7** | **58.6** | 58.7 | **60.8** | **66.6** | **65.4** | **69.9** | **68.4** | **74.2** | 78.6 | 68.0 | **71.3** | 74.1 | 71.6 | **66.8**±0.02 | +1.7 |
| EATA | 55.2 | 55.9 | 56.5 | 54.0 | 54.9 | 61.9 | 58.8 | 61.9 | 60.3 | 71.6 | 75.4 | 68.6 | 63.0 | 69.3 | 66.3 | 62.2±0.14 | +0.0 |
| + DEM* | 56.2 | 56.7 | 56.9 | 55.6 | 58.5 | 63.6 | 63.1 | 67.1 | 64.9 | 71.3 | 76.7 | 65.6 | 68.2 | 71.7 | 69.3 | 64.4±0.30 | +2.2 |
| + AdaDEM | 56.9 | 57.9 | 58.0 | 57.5 | 58.8 | 63.8 | 63.8 | 68.0 | 66.3 | 72.7 | 77.0 | 66.6 | 69.4 | 72.6 | 70.0 | 65.3±0.11 | +3.1 |
| DeYO | 53.3 | 54.4 | 54.3 | 55.1 | 55.1 | 61.9 | 55.0 | 64.3 | 63.1 | 71.7 | 77.2 | 67.2 | 65.8 | 71.5 | 68.4 | 62.6±0.32 | +0.0 |
| + DEM* | 56.4 | 57.2 | 57.4 | 59.1 | 58.8 | 64.3 | 59.8 | 67.3 | 66.8 | 74.1 | **79.3** | 70.7 | 67.9 | 73.8 | 70.9 | 65.6±0.03 | +3.0 |
| + AdaDEM | 53.6 | 54.7 | 54.7 | 54.5 | 56.2 | 62.1 | 54.4 | 63.2 | 63.5 | 71.7 | 77.1 | 67.5 | 65.2 | 71.7 | 68.9 | 62.6±0.10 | +0.0 |
| SAR | 51.0 | 51.6 | 52.4 | 52.4 | 49.5 | 56.2 | 49.2 | 21.7 | 43.9 | 66.6 | 73.2 | 66.5 | 51.4 | 64.4 | 63.3 | 54.2±0.07 | +0.0 |
| + DEM* | 51.1 | 51.1 | 52.8 | 53.4 | 48.5 | 56.4 | 48.9 | 54.8 | 56.7 | 67.8 | 75.7 | 67.4 | 52.9 | 66.3 | 64.0 | 57.9±0.04 | +3.7 |
| + AdaDEM | 56.6 | 57.8 | 57.6 | 58.5 | 60.0 | 64.1 | 62.2 | 68.4 | 66.5 | 73.3 | 78.9 | 66.9 | 69.0 | 74.1 | 71.5 | 65.7±0.07 | +11.5 |

replace the classical EM with AdaDEM, such as Tent, ETA, EATA, DeYO, SAR, TPT, Ent. Min., VAT, MinEnt, and *etc.*. Otherwise, we add AdaDEM, calculated using only unlabeled samples, to the original objective function, such as MixMatch, FixMatch, FreeMatch, AdvEnt, and *etc.*. Since Entropy Maximization is applied in RL to encourage exploration, we replace the implementation of conditional entropy in PPO with our AdaDEM while keeping the maximization strategy unchanged.

**Test-Time Adaptation Methods.** We compare with several TTA methods, including Tent [5], ETA [8], EATA [8], DeYO [11], SAR [29], and TPT [30]. We use ResNet50 and ViT-B/16 pre-trained on ImageNet-1K as the source models, except for using CLIP-RN50 and CLIP-ViT-B/16 for test-time prompt tuning. We follow the original methods' hyperparameter settings unless otherwise specified.

**Semi-Supervised Learning Methods.** We adopt VAT [3], MixMatch [24], FixMatch [23], and FreeMatch [22] implemented in USB [56] as the base algorithms. Meanwhile, we implement Ent. Min. as a baseline. Ent. Min. calculates the cross-entropy loss for labeled samples and the entropy minimization loss for unlabeled samples based on the work [1], where EM is weighted by $0.3$. ViT-S/2-32px is applied for CIFAR and EuroSat, ViT-S/16-224px for Semi-Aves, ViT-B/16-96px for STL-10, and ViT-T/2-32px for TissueMNIST. We follow the hyperparameter settings of these methods unless otherwise specified.

**Unsupervised Domain Adaptation Methods.** We mainly follow the work [27] to implement unsupervised domain adaptation in semantic segmentation. We use Deeplab-V2 [54] as the backbone. We employ MinEnt [27] and AdvEnt [27] as baselines and follow the hyperparameter settings of the original methods unless otherwise specified.

**Reinforcement Learning Methods.** We adopt Proximal Policy Optimization (PPO) [32] as the baseline. In PPO, Entropy Maximization is used as part of the objective function, which is weighted by $0.001$ by default. We replace the calculation method of conditional entropy in PPO with our AdaDEM while keeping the entropy-maximization strategy unchanged. We follow the hyperparameter setup implemented in RL Baselines3 Zoo unless otherwise specified.

# C More Experimental Results

## C.1 Experiments on Single-Domain & Continual Test-Time Adaptation Tasks

For single-domain and continual TTA tasks, we compare with previous state-of-the-art (SOTA) TTA methods, including Tent, ETA, EATA, DeYO, and SAR. We replace the classical EM loss function in these methods with our proposed DEM* and AdaDEM. As shown in Tables 5 and 6, we employ the ViT-B/16 pretrained on ImageNet-1K as the source model. Compared with the official Tent which

Table 6: Detailed experimental results on continual TTA task. Top-1 accuracy (%) is reported. We highlight the highest accuracy in **bold** and the second best as underline. Δ denotes the performance improvement relative to the baselines.

| Methods | Time Gauss | Noise Shot | Impul | Defcs | Blur Gls | Mtn | Zm | Snw | Weather Frst | Fg | Brt | Cnt | Digital Els | Px | Jpg | Mean | Δ |
|---|---|---|---|---|---|---|---|---|---|---|---|---|---|---|---|---|---|
| NoAdapt | 29.1 | 29.6 | 31.6 | 31.2 | 25.1 | 39.3 | 31.5 | 24.6 | 30.2 | 54.3 | 64.5 | 48.4 | 34.2 | 52.5 | 55.1 | $38.8_{\pm0.00}$ | - |
| Tent[†] | 49.4 | 54.2 | 56.3 | 46.9 | 48.1 | 56.7 | 52.0 | 55.7 | 61.0 | 68.2 | 77.2 | 64.5 | 52.8 | 68.1 | 67.5 | $58.6_{\pm0.09}$ | +0.0 |
| + DEM* | 50.8 | 57.1 | 59.3 | 56.4 | 58.7 | 62.6 | 62.0 | 65.9 | 66.2 | 72.0 | 76.6 | 66.6 | 66.5 | 71.4 | 69.1 | $64.1_{\pm0.05}$ | +5.5 |
| + AdaDEM | 54.9 | 59.2 | 60.1 | 53.3 | 56.9 | 61.7 | 58.6 | 63.3 | 65.3 | 70.9 | 77.6 | 64.3 | 61.7 | 71.3 | 69.3 | $63.2_{\pm0.16}$ | +4.6 |
| Tent | 51.6 | 56.2 | 57.5 | 49.2 | 51.2 | 57.5 | 53.2 | 56.0 | 61.2 | 68.2 | 77.2 | 64.8 | 20.9 | 1.6 | 1.8 | $48.5_{\pm0.71}$ | +0.0 |
| + DEM* | 53.1 | 58.7 | 59.5 | 56.2 | 58.2 | 63.1 | 63.5 | 66.1 | 65.9 | 72.1 | 76.4 | 66.6 | 67.3 | 71.7 | 69.1 | $64.5_{\pm0.14}$ | **+16.0** |
| + AdaDEM | 54.7 | 59.7 | 60.4 | 55.9 | 58.1 | 62.6 | 61.5 | 65.5 | 66.4 | 72.3 | 77.1 | 66.5 | 64.9 | 71.7 | 69.5 | $64.4_{\pm0.02}$ | +15.9 |
| ETA | 56.2 | 60.0 | 60.6 | 55.5 | 58.8 | 61.9 | 60.7 | 65.1 | 65.8 | 70.6 | 78.0 | 61.9 | 66.5 | 72.0 | 69.4 | $64.2_{\pm0.04}$ | +0.0 |
| + DEM* | 57.0 | 60.8 | **61.0** | 57.0 | 59.9 | 63.6 | 62.3 | 66.5 | 67.7 | 72.8 | 78.5 | 66.9 | 68.0 | 72.9 | 70.7 | $65.7_{\pm0.04}$ | +1.5 |
| + AdaDEM | 57.0 | **61.0** | 60.8 | 57.0 | **60.0** | 64.3 | 65.1 | **67.7** | 67.4 | 73.2 | 77.3 | 66.8 | 69.8 | 73.2 | 70.6 | $66.1_{\pm0.01}$ | +1.9 |
| EATA | 55.2 | 58.9 | 59.8 | 56.7 | 58.8 | 63.1 | 61.4 | 65.9 | 67.5 | 72.6 | 78.6 | 65.5 | 66.5 | 72.2 | 71.2 | $64.9_{\pm0.08}$ | +0.0 |
| + DEM* | 56.3 | 60.3 | 60.7 | **57.4** | **60.0** | 64.2 | 62.9 | 67.6 | **68.6** | **73.8** | 78.9 | **68.5** | 68.4 | 73.3 | **71.6** | $66.2_{\pm0.07}$ | +1.3 |
| + AdaDEM | **57.0** | 60.7 | 60.3 | 57.2 | **60.3** | **65.4** | **66.1** | 69.3 | 68.5 | 73.6 | 77.9 | 63.7 | **70.7** | **73.6** | 71.1 | $66.4_{\pm0.04}$ | +1.5 |
| DeYO | 53.3 | 56.6 | 56.5 | 44.9 | 53.0 | 56.2 | 22.9 | 59.0 | 60.2 | 67.4 | 76.2 | 57.5 | 62.9 | 69.9 | 67.1 | $57.6_{\pm0.36}$ | +0.0 |
| + DEM* | 56.2 | 60.3 | 60.4 | 57.3 | 60.0 | 63.8 | 56.4 | 66.2 | 67.4 | 73.3 | **79.0** | 67.7 | 67.9 | 73.4 | 70.9 | $65.4_{\pm0.12}$ | +7.8 |
| + AdaDEM | 51.5 | 54.5 | 55.3 | 50.7 | 53.2 | 57.0 | 50.9 | 57.9 | 60.8 | 64.5 | 75.6 | 60.5 | 59.2 | 67.6 | 66.3 | $59.0_{\pm0.05}$ | +1.4 |
| SAR | 51.0 | 54.2 | 55.2 | 50.5 | 52.5 | 56.5 | 52.8 | 50.8 | 41.4 | 66.7 | 75.5 | 64.2 | 52.9 | 65.1 | 65.6 | $57.0_{\pm0.05}$ | +0.0 |
| + DEM* | 51.4 | 57.1 | 58.9 | 53.6 | 54.6 | 60.1 | 56.3 | 62.3 | 65.0 | 69.9 | 78.3 | 67.2 | 61.1 | 70.6 | 70.1 | $62.4_{\pm0.03}$ | +5.4 |
| + AdaDEM | 54.4 | 58.5 | 58.6 | 53.4 | 57.5 | 61.2 | 59.0 | 62.9 | 64.0 | 69.8 | 77.8 | 63.1 | 64.4 | 71.3 | 68.8 | $63.0_{\pm0.05}$ | +6.0 |

Table 7: Detailed experimental results on test-time prompt tuning task. Top-1 accuracy (%) is reported. OOD Avg. is average accuracies on ImageNet variants, including -A, -V2., -R., and -S..

| Methods | ImageNet-1K | ImageNet-A | ImageNet-V2. | ImageNet-R. | ImageNet-S. | Average | Δ | OOD Avg. | Δ |
|---|---|---|---|---|---|---|---|---|---|
| | | | *CLIP-RN50* | | | | | | |
| Zero-Shot | $58.2_{\pm0.00}$ | $21.8_{\pm0.00}$ | $51.4_{\pm0.00}$ | $56.2_{\pm0.00}$ | $33.4_{\pm0.00}$ | $44.2_{\pm0.00}$ | +0.0 | $40.7_{\pm0.00}$ | +0.0 |
| Ensemble | $59.8_{\pm0.00}$ | $23.2_{\pm0.00}$ | $52.9_{\pm0.00}$ | **$60.7_{\pm0.00}$** | $35.5_{\pm0.00}$ | $46.4_{\pm0.00}$ | +2.2 | $43.1_{\pm0.00}$ | +2.4 |
| TPT | $60.7_{\pm0.07}$ | $26.1_{\pm0.10}$ | $54.6_{\pm0.02}$ | $58.9_{\pm0.08}$ | $35.2_{\pm0.09}$ | $47.1_{\pm0.06}$ | +2.9 | $43.7_{\pm0.05}$ | +3.0 |
| + DEM* | $61.3_{\pm0.09}$ | $25.5_{\pm0.07}$ | $55.0_{\pm0.10}$ | $59.7_{\pm0.12}$ | $35.6_{\pm0.08}$ | $47.4_{\pm0.04}$ | +3.2 | $43.9_{\pm0.06}$ | +3.2 |
| + AdaDEM | $60.7_{\pm0.04}$ | $29.2_{\pm0.19}$ | $54.8_{\pm0.22}$ | $58.8_{\pm0.05}$ | $35.4_{\pm0.03}$ | $47.8_{\pm0.07}$ | +3.6 | $44.5_{\pm0.09}$ | +3.8 |
| CoOp | $63.3_{\pm0.00}$ | $23.1_{\pm0.00}$ | $55.4_{\pm0.00}$ | $56.6_{\pm0.00}$ | $34.7_{\pm0.00}$ | $46.6_{\pm0.00}$ | +2.4 | $42.4_{\pm0.00}$ | +1.7 |
| TPT (CoOp) | $65.4_{\pm0.06}$ | $28.9_{\pm0.14}$ | $58.2_{\pm0.10}$ | $59.0_{\pm0.09}$ | $36.3_{\pm0.15}$ | $49.6_{\pm0.07}$ | +5.4 | $45.6_{\pm0.07}$ | +4.9 |
| + AdaDEM | **$65.6_{\pm0.05}$** | **$31.3_{\pm0.10}$** | **$58.5_{\pm0.22}$** | $59.3_{\pm0.10}$ | **$36.3_{\pm0.11}$** | **$50.2_{\pm0.06}$** | **+6.0** | **$46.4_{\pm0.06}$** | **+5.7** |
| | | | *CLIP-ViT-B/16* | | | | | | |
| Zero-Shot | $66.7_{\pm0.00}$ | $47.9_{\pm0.00}$ | $60.9_{\pm0.00}$ | $74.0_{\pm0.00}$ | $46.1_{\pm0.00}$ | $59.1_{\pm0.00}$ | +0.0 | $57.2_{\pm0.00}$ | +0.0 |
| Ensemble | $68.3_{\pm0.00}$ | $49.9_{\pm0.00}$ | $61.9_{\pm0.00}$ | $77.7_{\pm0.00}$ | $48.2_{\pm0.00}$ | $61.2_{\pm0.00}$ | +2.1 | $59.4_{\pm0.00}$ | +2.2 |
| TPT | $69.0_{\pm0.04}$ | $54.5_{\pm0.09}$ | $63.4_{\pm0.13}$ | $77.0_{\pm0.06}$ | $48.0_{\pm0.13}$ | $62.4_{\pm0.05}$ | +3.3 | $60.7_{\pm0.06}$ | +3.5 |
| + DEM* | $68.9_{\pm0.03}$ | $54.8_{\pm0.09}$ | $63.5_{\pm0.11}$ | $77.1_{\pm0.08}$ | $47.9_{\pm0.06}$ | $62.5_{\pm0.06}$ | +3.4 | $60.8_{\pm0.08}$ | +3.6 |
| + AdaDEM | $69.4_{\pm0.12}$ | $58.8_{\pm0.18}$ | $64.0_{\pm0.06}$ | $77.6_{\pm0.21}$ | $48.6_{\pm0.05}$ | $63.7_{\pm0.05}$ | +4.6 | $62.2_{\pm0.09}$ | +5.0 |
| CoOp | $71.5_{\pm0.00}$ | $49.7_{\pm0.00}$ | $64.2_{\pm0.00}$ | $75.2_{\pm0.00}$ | $48.0_{\pm0.00}$ | $61.7_{\pm0.00}$ | +2.6 | $59.3_{\pm0.00}$ | +2.1 |
| TPT (CoOp) | $73.6_{\pm0.05}$ | $57.9_{\pm0.12}$ | $66.9_{\pm0.08}$ | $77.2_{\pm0.04}$ | $49.2_{\pm0.07}$ | $64.9_{\pm0.06}$ | +5.8 | $62.8_{\pm0.06}$ | +5.6 |
| + AdaDEM | **$73.7_{\pm0.07}$** | **$60.3_{\pm0.11}$** | **$66.9_{\pm0.19}$** | **$77.9_{\pm0.14}$** | **$49.3_{\pm0.07}$** | **$65.6_{\pm0.02}$** | **+6.5** | **$63.6_{\pm0.04}$** | **+6.4** |

applies SGD with momentum, our implemented Tent[†] utilizes vanilla SGD as the optimizer and achieves better performance on both tasks. However, SGD with momentum has higher potential, and applying DEM* and AdaDEM can bring higher performance gains. ETA, EATA, SAR, and DeYO adopt sample-selection-based methods to screen high-confident test samples for model optimization. These methods assign a higher weight to high-certainty test samples. This re-weighting approach partially alleviates the reward collapse phenomenon in the classical EM. Nevertheless, there is still room for improvement, which can be observed from the performance gains brought by DEM*. EATA and SAR use the Fisher information of in-distribution samples from the source model and the sharpness-aware optimizer respectively to stabilize the model's optimization in the dynamically changing environment. Our DEM* and AdaDEM can be directly applied to these prior-based methods and achieve significant performance improvements.

## C.2 Experiments on Test-Time Prompt Tuning Task

Test-time prompt tuning is an episodic TTA task. It focuses on optimizing the learnable text prompts of the source model using a single test sample and making predictions for the current batch with

Table 8: Detailed experimental results (a) on semi-supervised learning. Top-1 accuracy (%) is reported. We highlight the highest accuracy in **bold** and the second best as underline. $\Delta$ denotes the performance improvement relative to the baselines.

| Methods | CIFAR-10 | | | | | CIFAR-100 | | | | | Semi-Aves | |
|---|---|---|---|---|---|---|---|---|---|---|---|---|
| | 4 | 25 | 400 | Mean | $\Delta$ | 2 | 4 | 25 | Mean | $\Delta$ | 15~53 | $\Delta$ |
| Ent. Min. | 93.2±5.7 | 97.5±0.2 | 98.5±0.0 | 96.4±1.9 | +0.0 | 58.8±2.6 | 74.6±1.1 | 84.4±0.2 | 72.6±0.5 | +0.0 | 59.9±0.7 | +0.0 |
| + AdaDEM | 95.7±0.6 | 97.4±0.1 | 98.5±0.0 | 97.2±0.2 | +0.8 | 66.2±1.6 | 76.3±0.8 | 85.1±0.0 | 75.8±0.3 | **+3.2** | 61.0±0.2 | **+1.1** |
| Vat (w/ Ent. Min.) | 94.7±6.0 | 98.6±0.0 | 98.9±0.0 | 97.4±2.0 | +0.0 | 68.4±2.1 | 78.3±0.5 | 85.8±0.4 | 77.5±0.7 | +0.0 | 61.0±0.4 | +0.0 |
| + AdaDEM | 97.8±0.2 | 98.7±0.1 | 98.8±0.0 | 98.5±0.1 | **+1.1** | 70.8±0.2 | 79.2±1.1 | 86.3±0.4 | 78.8±0.7 | +1.3 | 61.8±0.0 | +0.8 |
| MixMatch | 98.1±0.7 | 98.5±0.1 | 98.9±0.1 | 98.5±0.3 | +0.0 | 62.8±0.7 | 73.6±0.6 | 84.9±0.2 | 73.8±0.4 | +0.0 | 62.6±0.2 | +0.0 |
| + AdaDEM | 97.5±1.5 | 98.6±0.1 | 98.9±0.1 | 98.3±0.6 | -0.2 | 64.1±0.7 | 74.6±0.2 | 85.7±0.2 | 74.8±0.2 | +1.0 | 62.5±0.1 | -0.1 |
| FixMatch | 97.5±2.0 | 98.8±0.1 | 99.1±0.0 | 98.4±0.7 | +0.0 | 69.9±1.7 | 80.6±0.6 | 87.2±0.2 | 79.3±0.6 | +0.0 | **68.2±0.3** | +0.0 |
| + AdaDEM | 98.4±0.4 | 98.7±0.0 | 99.1±0.0 | 98.7±0.2 | +0.3 | 71.3±2.3 | 80.1±0.3 | 87.0±0.2 | 79.5±0.9 | +0.2 | 67.9±0.1 | -0.3 |
| FreeMatch | 98.7±0.1 | 99.0±0.1 | 99.1±0.0 | 98.9±0.0 | +0.0 | 76.5±1.3 | 83.6±1.0 | 87.5±0.3 | 82.6±0.1 | +0.0 | 67.0±0.3 | +0.0 |
| + AdaDEM | **98.8±0.2** | **99.0±0.0** | **99.1±0.0** | **99.0±0.1** | +0.1 | **77.5±0.0** | **84.3±0.1** | **87.7±0.2** | **83.1±0.1** | +0.5 | 67.3±0.2 | +0.3 |

Table 9: Detailed experimental results (b) on semi-supervised learning. Top-1 accuracy (%) is reported. We highlight the highest accuracy in **bold** and the second best as underline. $\Delta$ denotes the performance improvement relative to the baselines.

| Methods | STL-10 | | | | EuroSat | | | | TissueMNIST | | | |
|---|---|---|---|---|---|---|---|---|---|---|---|---|
| | 4 | 10 | Mean | $\Delta$ | 2 | 4 | Mean | $\Delta$ | 10 | 50 | Mean | $\Delta$ |
| Ent. Min. | 76.3±3.0 | 88.5±0.6 | 82.4±1.6 | +0.0 | 66.6±5.7 | 86.4±2.5 | 76.5±3.6 | +0.0 | 44.6±3.7 | 50.0±1.7 | 47.3±2.6 | +0.0 |
| + AdaDEM | 80.4±0.7 | 89.2±0.1 | 84.8±0.3 | **+2.4** | 76.4±1.2 | 91.0±0.3 | 83.7±0.8 | **+7.2** | 46.8±1.9 | 51.8±0.7 | 49.3±1.3 | +2.0 |
| Vat (w/ Ent. Min) | 81.6±1.8 | 89.3±0.6 | 85.4±0.9 | +0.0 | 81.9±9.9 | 91.3±2.9 | 86.6±5.4 | +0.0 | 41.6±7.9 | 48.7±1.9 | 45.2±4.8 | +0.0 |
| + AdaDEM | 83.8±0.2 | 89.9±0.1 | 86.9±0.1 | +1.5 | 90.3±1.8 | 92.1±0.1 | 91.2±0.9 | +4.6 | **48.0±1.3** | 50.8±0.1 | 49.4±0.6 | **+4.2** |
| MixMatch | 76.7±3.1 | 89.2±1.1 | 82.9±2.1 | +0.0 | 72.0±5.6 | 86.6±7.2 | 79.3±4.3 | +0.0 | 44.7±2.3 | 51.3±1.8 | 48.0±1.7 | +0.0 |
| + AdaDEM | 80.2±0.1 | 90.2±0.1 | 85.2±0.1 | +2.3 | 76.4±0.2 | 91.4±4.9 | 83.9±2.4 | +4.6 | 46.8±1.7 | 53.2±0.2 | **50.0±0.9** | +2.0 |
| FixMatch | 84.0±2.6 | 93.1±0.5 | 88.5±1.2 | +0.0 | 87.7±7.2 | 96.2±0.1 | 91.9±3.6 | +0.0 | 44.6±4.8 | 48.8±2.1 | 46.7±3.3 | +0.0 |
| + AdaDEM | 83.8±0.4 | 92.1±1.9 | 87.9±0.8 | -0.6 | **96.0±1.0** | **97.8±0.4** | **96.9±0.7** | +5.0 | 46.5±1.7 | **53.4±0.9** | 49.9±1.3 | +3.2 |
| FreeMatch | 86.9±1.8 | 92.5±1.5 | 89.7±1.6 | +0.0 | 95.2±1.7 | 96.4±0.5 | 95.8±1.0 | +0.0 | 42.8±4.6 | 48.3±2.0 | 45.6±3.3 | +0.0 |
| + AdaDEM | **89.6±0.7** | **93.7±0.3** | **91.6±0.2** | +1.9 | 95.6±0.1 | 96.2±0.0 | 95.9±0.1 | +0.1 | 46.3±0.0 | 49.6±0.8 | 47.9±0.4 | +2.3 |

the updated parameters. It emphasizes the ability to learn rapidly and adapt extremely well to a single test sample, often employing aggressive optimization strategies such as more complex image augmentation functions, larger learning rates, and more test-time adaptation steps. We adopt CLIP-RN50 and CLIP-ViT-B/16 as the source models. As shown in the Table. 7, CLIP models' zero-shot prediction and the experimental results based on a prompt ensemble method [53] are employed as baselines. We compare with TPT, a widely used method for test-time prompt tuning, and CoOp, a prompt-tuning method for supervised few-shot learning. We modify the loss function of TPT to our proposed DEM* and AdaDEM. We find similar conclusions in other TTA tasks, that is, DEM* and AdaDEM improve the optimization performance by addressing the potential limitations in the classical EM. They enable the source model to adapt in aggressive optimization strategies with large learning rates and multiple steps for TTA.

## C.3 Experiments on Semi-Supervised Learning

Entropy Minimization is widely used in semi-supervised learning (SSL) tasks. A common prior assumption for SSL is the low-density separation between classes. EM reduces the overlap of models' output probability distribution by decreasing the conditional entropy of the data, causing the density of data points to be lower at the decision boundary. To verify the effectiveness of our proposed AdaDEM in SSL tasks, we conduct experiments on six common SSL benchmarks. The results are presented in Table 8 and 9. For each benchmark, we set different numbers of available labeled samples $N_l$ per class for training. Specifically, we set $N_l = 4/25/400$ for CIFAR-10, $N_l = 2/4/25$ for CIFAR-100, $N_l = 4/10$ for STL-10, $N_l = 2/4$ for EuroSat, $N_l = 10/50$ for TissueMNIST, and $N_l = 15 \sim 53$ for Semi-Aves. We report the average top-1 classification accuracy under different numbers of labeled samples. We replace the classical EM in Ent. Min. [1] and VAT (w/ Ent. Min.) [3] with AdaDEM, and introduce AdaDEM as a part of the loss functions in MixMatch [24], FixMatch [23], and FreeMatch [22] which do not use Entropy Minimization. Through experiments, we demonstrate that AdaDEM outperforms the classical EM. Meanwhile, incorporating AdaDEM into existing SSL methods can effectively improve the performance of the original algorithms.

Table 10: Detail experimental results on unsupervised domain adaptation in semantic segmentation. The standard mIoU is reported. We highlight the highest accuracy in **bold** and the second best as underline. Δ denotes the performance improvement relative to the baselines.

| Methods | road | sidewalk | building | wall | fence | pole | light | sign | veg | terrain | sky | person | rider | car | truck | bus | train | mbike | bike | mIoU | Δ |
|---|---|---|---|---|---|---|---|---|---|---|---|---|---|---|---|---|---|---|---|---|---|
| MinEnt | 84.4 | 24.3 | 77.4 | 22.1 | 21.3 | 26.5 | 33.1 | 19.2 | 82.7 | 30.9 | 76.4 | 58.0 | 26.6 | 75.4 | 31.4 | 38.0 | 2.0 | 25.5 | 35.5 | 41.6±0.47 | +0.0 |
| + AdaDEM | 85.5 | 22.6 | 78.8 | 21.4 | 24.6 | 27.2 | **34.4** | 19.6 | 82.5 | 28.8 | 78.0 | 58.5 | 28.5 | 80.1 | 34.4 | 41.2 | 1.8 | 24.5 | **38.1** | 42.7±0.13 | +1.1 |
| AdvEnt | **89.1** | 25.4 | **81.4** | 26.7 | 25.6 | **29.4** | 32.8 | **22.5** | 83.8 | 35.1 | 77.5 | 57.9 | 28.8 | 84.3 | 30.3 | 41.8 | 0.9 | **29.3** | 26.0 | 43.6±0.19 | +0.0 |
| + DEM* | 89.0 | **32.2** | 80.9 | **28.3** | 25.4 | 28.4 | 33.7 | 20.9 | 84.1 | **35.6** | 77.8 | 58.8 | 29.1 | 83.6 | 34.7 | 41.3 | 1.6 | 29.1 | 32.6 | 44.6±0.32 | +1.0 |
| + AdaDEM | 88.9 | 30.4 | 81.2 | 27.9 | **27.0** | 28.7 | 34.0 | 21.0 | 84.1 | 35.4 | 78.6 | 59.1 | 29.8 | 84.4 | 36.4 | 42.6 | 1.6 | 28.0 | 33.4 | **44.9±0.17** | **+1.3** |

Figure 7: Visualization of the pixel prediction results and pixel entropy of baselines and AdaDEM.

## C.4 Experiments on Unsupervised Domain Adaptation

Another role of Entropy Minimization is to bridge the domain gap between the source training distribution and the target test distribution. The classical EM pushes the decision boundaries of DNNs towards the low-density regions of the target distribution. On the other hand, conditional entropy is also a widely validated calibration tool for DNNs, which measures the prediction error and distribution shifts to some extent. Vu *et al.* vu2019advent demonstrate that the semantic segmentation model trained in the source distribution has high prediction entropy and blurred boundaries for objects in out-of-distribution images. We verify the effectiveness of DEM* and AdaDEM in the semantic segmentation task, and use MinEnt and AdvEnt [27] as baselines for comparison. The experimental results are shown in Table 10. We also visualize the segmentation results of the baselines and AdaDEM, as shown in Fig. 7. The experimental results show that compared with the classical EM, AdaDEM can effectively improve the accuracy of semantic segmentation, especially for tail classes, and avoid DNNs' tendency to predict dominant classes. At the same time, AdaDEM can effectively reduce the prediction entropy of DNNs for image pixels, resulting in clearer object boundaries and thus bridging the domain gap between the source and target distributions.

## C.5 Experiments on Reinforcement Learning

For reinforcement learning, Entropy Minimization could increase the certainty of policies, thereby limiting the exploration of agents. Therefore, reinforcement learning methods commonly use the

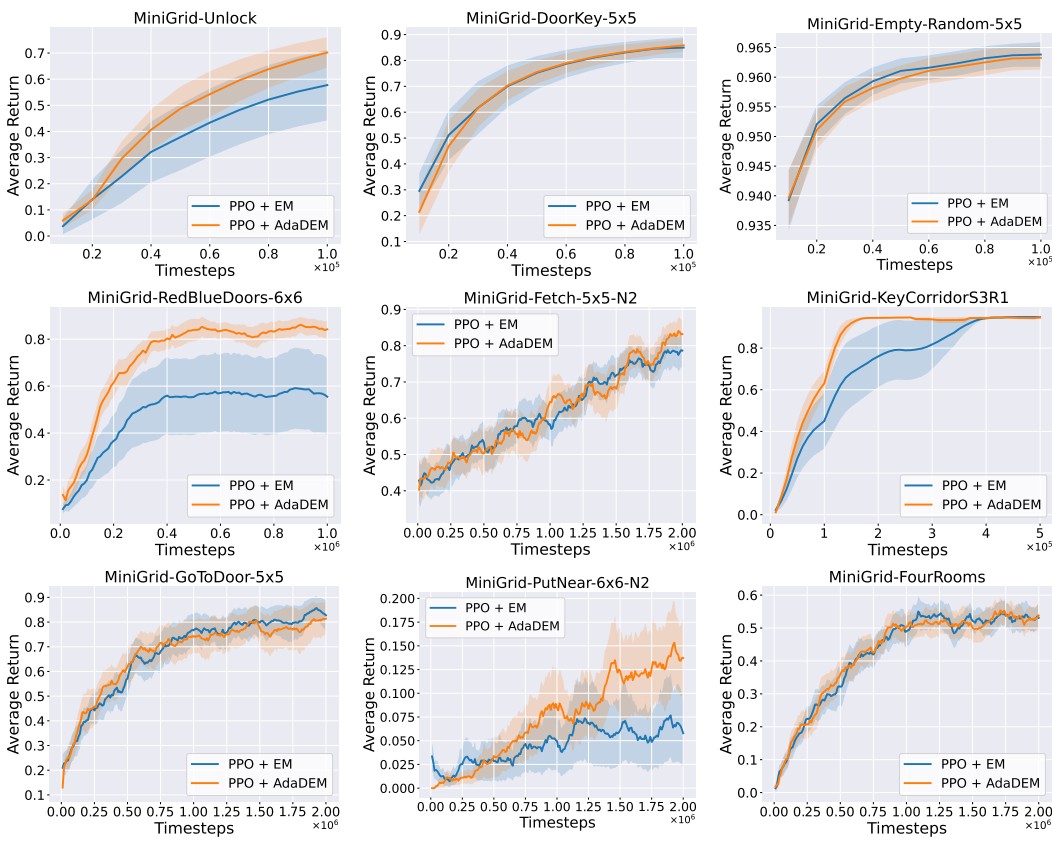

Figure 8: The curves of average return with respect to timesteps in 9 environments of Minigrid for Reinforcement Learning.

Entropy Maximization strategy to prevent premature convergence and encourage exploration. We aim to prove that AdaDEM is not limited to the Entropy Minimization strategy and is also applicable to the Entropy Maximization strategy in reinforcement learning. We adopt Proximal Policy Optimization (PPO) as the baseline and compare the experimental results of adding the classical EM and our AdaDEM to PPO, as shown in Fig. 8. We keep the Entropy Maximization strategy unchanged. The experimental results demonstrate that replacing the classical EM with AdaDEM can effectively improve or maintain a comparable performance of reinforcement learning methods.

# D   Additional Discussions

## D.1   Effect of Temperature $\tau$ in DEM*

As stated in Sec. 3.2, $\tau$ controls the contribution of samples with different certainties to model optimization by reshaping the reward curve. As shown in Fig. 3 (left), the larger the value of $\tau$ is, the higher the reward given to high-confident samples than to low-confident ones, which makes the model more inclined to learn high-confident samples. On the contrary, the lower the value of $\tau$ is, the more the model tends to learn low-confident samples. When DNNs classify all samples accurately, a lower value of $\tau$ promotes the model to rapidly improve the prediction probability of samples and accelerate the model optimization. However, poorly calibrated DNNs often have very low accuracy on out-of-distribution samples. In this case, a low value of $\tau$ leads to incorrect model optimization. Therefore, setting $\tau$ to a value slightly larger than $1.0$ enables DNNs to stably and effectively learn from data distributions with low certainty (distributions that deviate significantly from the source training distribution). Additionally, as shown in Fig. 3 (center), we find that the best value of $\tau$ is positively correlated with the average prediction probability of DNNs for samples in a specific distribution. That is, when the model has a higher average prediction accuracy for a certain data distribution, increasing the value of $\tau$ maximizes the efficiency of DNNs in learning this data distribution within a limited optimization time.

Table 11: Optimal hyperparameters $(\tau^*, \alpha^*)$ of DEM* applied to different methods on various tasks, and the resulting performance improvement brought by DEM*.

| Methods | Tent† (RN) | + DEM* | Tent† (ViT) | + DEM* | ETA | + DEM* | EATA | + DEM* | DeYO | + DEM* | SAR | + DEM* |
|---|---|---|---|---|---|---|---|---|---|---|---|---|
| | | | | | Single-Domain Test-Time Adaptation | | | | | | | |
| $\tau^*$ | 1.0 | 1.1 | 1.0 | 1.3 | 1.0 | 1.2 | 1.0 | 1.3 | 1.0 | 1.1 | 1.0 | 1.0 |
| $\alpha^*$ | 1.0 | 0.0 | 1.0 | 1.8 | 1.0 | 1.2 | 1.0 | 0.2 | 1.0 | 1.6 | 1.0 | 2.0 |
| Acc. | $40.0_{\pm0.03}$ | $41.8_{\pm0.05}$ | $53.1_{\pm0.65}$ | $56.0_{\pm0.32}$ | $65.1_{\pm0.10}$ | $66.3_{\pm0.04}$ | $62.2_{\pm0.14}$ | $64.4_{\pm0.30}$ | $62.6_{\pm0.32}$ | $65.6_{\pm0.03}$ | $54.2_{\pm0.07}$ | $57.9_{\pm0.04}$ |
| $\Delta$ | +0.0 | +1.8 | +0.0 | +2.9 | +0.0 | +1.2 | +0.0 | +2.2 | +0.0 | +3.0 | +0.0 | +3.7 |
| | | | | | Continual Test-Time Adaptation | | | | | | | |
| $\tau^*$ | 1.0 | 1.4 | 1.0 | 0.8 | 1.0 | 1.1 | 1.0 | 1.3 | 1.0 | 0.9 | 1.0 | 1.0 |
| $\alpha^*$ | 1.0 | 1.4 | 1.0 | 1.0 | 1.0 | 1.2 | 1.0 | 1.2 | 1.0 | 1.4 | 1.0 | 1.8 |
| Acc. | $31.2_{\pm0.11}$ | $39.0_{\pm0.02}$ | $58.6_{\pm0.09}$ | $64.1_{\pm0.05}$ | $64.2_{\pm0.04}$ | $65.7_{\pm0.04}$ | $64.9_{\pm0.08}$ | $66.2_{\pm0.07}$ | $57.6_{\pm0.36}$ | $65.4_{\pm0.12}$ | $57.0_{\pm0.05}$ | $62.4_{\pm0.03}$ |
| $\Delta$ | +0.0 | +7.8 | +0.0 | +5.5 | +0.0 | +1.5 | +0.0 | +1.3 | +0.0 | +7.8 | +0.0 | +5.4 |

| Methods | CLIP-RN50 | TPT | | + DEM* | CLIP-ViT-B/16 | TPT | | + DEM* |
|---|---|---|---|---|---|---|---|---|
| | | | Test-Time Prompt Tuning | | | | | |
| $\tau^*$ | - | 1.0 | | 1.0 | - | 1.0 | | 0.8 |
| $\alpha^*$ | - | 1.0 | | 0.6 | - | 1.0 | | 1.0 |
| Acc. | $44.2_{\pm0.00}$ | $47.1_{\pm0.06}$ | | $47.4_{\pm0.04}$ | $59.1_{\pm0.00}$ | $62.4_{\pm0.05}$ | | $62.5_{\pm0.06}$ |
| $\Delta$ | +0.0 | +2.9 | | +3.2 | +0.0 | +3.3 | | +3.4 |

## D.2 Effect of Weight $\alpha$ in DEM*

In GMC, $\alpha$ suppresses the overfitting of DNNs to over-confident samples by scaling the penalty on logits. As shown in Fig. 4 (left), increasing the value of $\alpha$ causes DNNs to impose greater penalties on samples with prediction probabilities approaching $1.0$. For poorly calibrated DNNs, increasing the value of $\alpha$ reduces the disruption of over-confident samples to DNNs optimization. For data distributions with high prediction accuracy, decreasing the value of $\alpha$ promotes rapid model optimization and enhances the learning efficiency of high-confident samples. Therefore, a lower value of $\alpha$ is suitable for the aggressive optimization strategy, while a higher value of $\alpha$ is suitable for alleviating the catastrophic forgetting problem caused by DNNs overfitting. The ablation study in Fig. 4 (center) validates this point.

## D.3 Optimal Hyperparameters $(\tau^*, \alpha^*)$ for DEM*

In Table 11, we obtain the optimal hyperparameters $(\tau^*, \alpha^*)$ for different methods on various tasks through hyperparameter search by the fast TPE algorithm. For test-time prompt tuning, we employ well-calibrated CLIP models as the source models. It is suitable for leveraging an aggressive optimization strategy to enable CLIP models to quickly learn the target data distribution. Therefore, a lower value of $\tau$ or $\alpha$ is optimal. We mainly use ViT-B/16 for TTA. Hence, for single-domain TTA, a lower $\alpha$ and a higher $\tau$ are suitable for the rapid optimization of ViT-B/16 and enhance the contribution of high-confident samples to model optimization. For continual TTA, higher values of $\tau$ and $\alpha$ prevent ViT-B/16 from overfitting to a Single static distribution and improve the model's adaptability in dynamically changing environments and its ability to resist catastrophic forgetting.

## D.4 Ablation Studies of DEM and AdaDEM

In Tables 12 and 13, we report the detailed ablation experiment results of DEM and AdaDEM, respectively. Different from the official Tent [5], which uses SGD with Momentum as the optimizer, we employ vanilla SGD to verify the learning efficiency of TTA algorithms on real-time incoming test samples, while avoiding the influence of historical gradients on the learning of current samples. This implementation leads to better accuracies than the official Tent. In some cases, applying CADF alone improves the performance of DNNs on the single-domain TTA task. However, CADF suffers from catastrophic forgetting in dynamically changing environments. Therefore, DEM* searches the optimal hyperparameters $(\tau^*, \alpha^*)$ to stabilize the model optimization process and enhance the model's learning efficiency for the target distribution. AdaDEM-Norm solves the reward collapse phenomenon in the classical EM by introducing $\delta$, the L1-norm of the reward brought from CADF. It improves the performance of Tent. Meanwhile, AdaDEM-MEC uses the estimated prior label distribution in the marginal entropy calibrator to alleviate the model's overfitting to dominant classes, resulting in performance improvement on continual tasks. AdaDEM comprehensively utilizes AdaDEM-Norm and AdaDEM-MEC as alternatives to the classical EM and DEM. It avoids the additional

Table 12: Detailed ablation studies of DEM and AdaDEM in the single-domain TTA task. We highlight the highest accuracy in **bold** and the second best as underline. $\Delta$ denotes the performance improvement relative to the baselines.

| Methods | Noise | | | Blur | | | | Weather | | | | Digital | | | | Mean | $\Delta$ |
| | Gauss | Shot | Impul | Defcs | Gls | Mtn | Zm | Snw | Frst | Fg | Brt | Cnt | Els | Px | Jpg | | |
|---|---|---|---|---|---|---|---|---|---|---|---|---|---|---|---|---|---|
| | | | | | | | | *ResNet50* | | | | | | | | | |
| NoAdapt | 15.3 | 15.9 | 15.8 | 15.1 | 15.2 | 26.4 | 38.9 | 34.3 | 33.1 | 48.1 | 65.2 | 16.8 | 44.1 | 48.8 | 39.6 | 31.5±0.00 | - |
| EM (Tent) | 24.8 | 26.3 | 25.8 | 24.4 | 23.7 | 36.9 | 46.9 | 43.9 | 39.4 | 55.6 | 66.8 | 26.7 | 52.1 | 56.5 | 49.7 | 40.0±0.03 | +0.0 |
| CADF | 28.3 | 29.2 | 29.3 | 27.2 | 26.5 | 40.6 | 48.5 | 46.8 | 40.9 | 56.7 | 66.0 | 22.8 | 54.1 | 57.6 | 51.5 | 41.7±0.05 | +1.7 |
| DEM* | 28.3 | 29.2 | 29.3 | 27.1 | 26.5 | 40.7 | 48.5 | 46.8 | 41.0 | 56.8 | 66.0 | 23.2 | 54.2 | 57.6 | 51.5 | 41.8±0.05 | +1.8 |
| AdaDEM-Norm | 30.5 | 32.7 | 32.0 | 29.6 | 28.5 | 44.0 | 50.7 | 49.2 | 41.0 | 58.6 | 67.6 | 22.1 | 56.4 | 59.7 | 53.6 | 43.7±0.10 | +3.7 |
| AdaDEM-MEC | 31.7 | 34.2 | 33.1 | 30.3 | 29.6 | 44.8 | 50.0 | 49.5 | 41.8 | 57.9 | 66.0 | 30.0 | 55.6 | 58.6 | 53.0 | 44.4±0.04 | +4.4 |
| AdaDEM | 31.8 | 33.9 | 33.0 | 30.6 | 30.0 | 44.0 | 50.3 | 49.3 | 42.8 | 58.0 | 66.2 | 34.2 | 55.7 | 58.7 | 53.0 | 44.8±0.05 | +4.8 |
| | | | | | | | | *ViT-B/16* | | | | | | | | | |
| NoAdapt | 29.1 | 29.6 | 31.6 | 31.2 | 25.1 | 39.3 | 31.5 | 24.6 | 30.2 | 54.3 | 64.5 | 48.4 | 34.2 | 52.5 | 55.1 | 38.8±0.00 | - |
| EM (Tent) | 54.2 | 54.8 | 55.4 | 56.6 | 54.0 | 60.8 | 34.2 | 3.8 | 9.6 | 70.9 | 76.5 | 69.4 | 59.5 | 70.0 | 67.2 | 53.1±0.65 | +0.0 |
| CADF | 50.1 | 50.6 | 51.6 | 49.0 | 44.3 | 55.0 | 46.1 | 6.6 | 15.0 | 64.2 | 72.8 | 65.3 | 33.0 | 63.4 | 61.6 | 48.6±0.42 | -4.5 |
| DEM* | 51.7 | 51.9 | 53.3 | 54.8 | 51.2 | 58.7 | 50.8 | 14.7 | 44.3 | 69.9 | 77.0 | 68.5 | 57.3 | 69.0 | 66.8 | 56.0±0.32 | +2.9 |
| AdaDEM-Norm | 54.3 | 55.2 | 55.6 | 56.5 | 54.2 | 61.1 | 41.0 | 3.7 | 13.4 | 71.2 | 76.4 | 69.3 | 60.5 | 70.3 | 67.4 | 54.0±0.45 | +0.9 |
| AdaDEM-MEC | 56.0 | 57.1 | 57.0 | 41.4 | 59.1 | 64.1 | 61.2 | 2.7 | 62.5 | 73.4 | 77.0 | 50.2 | 67.9 | 71.9 | 69.7 | 58.1±0.23 | +5.0 |
| AdaDEM | 57.2 | 58.2 | 58.1 | 58.9 | 60.2 | 65.5 | 63.3 | 2.0 | 66.0 | 73.7 | 78.0 | 68.7 | 69.2 | 73.3 | 70.4 | 61.5±0.20 | +8.4 |

Table 13: Detailed ablation studies of DEM and AdaDEM in the continual TTA task. We highlight the highest accuracy in **bold** and the second best as underline. $\Delta$ denotes the performance improvement relative to the baselines.

| Methods | Time → | | | | | | | | | | | | | | | Mean | $\Delta$ |
| | Noise | | | Blur | | | | Weather | | | | Digital | | | | | |
| | Gauss | Shot | Impul | Defcs | Gls | Mtn | Zm | Snw | Frst | Fg | Brt | Cnt | Els | Px | Jpg | | |
|---|---|---|---|---|---|---|---|---|---|---|---|---|---|---|---|---|---|
| | | | | | | | | *ResNet50* | | | | | | | | | |
| NoAdapt | 15.3 | 15.9 | 15.8 | 15.1 | 15.2 | 26.4 | 38.9 | 34.3 | 33.1 | 48.1 | 65.2 | 16.8 | 44.1 | 48.8 | 39.6 | 31.5±0.00 | - |
| EM (Tent) | 24.8 | 32.9 | 32.7 | 24.2 | 25.9 | 30.2 | 37.7 | 30.2 | 28.3 | 36.4 | 49.2 | 17.7 | 32.5 | 35.3 | 30.0 | 31.2±0.11 | +0.0 |
| CADF | 18.1 | 24.2 | 26.6 | 22.6 | 25.7 | 33.8 | 44.1 | 36.9 | 36.4 | 47.2 | 59.9 | 26.3 | 47.1 | 49.3 | 43.8 | 36.1±0.03 | +4.9 |
| DEM* | 18.7 | 26.2 | 28.9 | 23.9 | 26.9 | 36.5 | 45.6 | 39.8 | 39.5 | 49.8 | 62.8 | 32.8 | 49.6 | 54.0 | 49.3 | 39.0±0.02 | +7.8 |
| AdaDEM-Norm | 18.9 | 25.2 | 27.1 | 22.6 | 25.6 | 34.3 | 44.6 | 38.0 | 37.9 | 48.7 | 62.2 | 28.4 | 49.3 | 52.8 | 47.4 | 37.5±0.05 | +6.3 |
| AdaDEM-MEC | 20.5 | 28.0 | 29.5 | 23.5 | 27.0 | 34.2 | 43.8 | 36.9 | 37.0 | 47.2 | 60.5 | 27.8 | 48.5 | 51.5 | 46.5 | 37.5±0.04 | +6.3 |
| AdaDEM | 20.5 | 28.3 | 29.9 | 23.8 | 27.5 | 34.5 | 44.0 | 36.9 | 36.9 | 47.4 | 60.3 | 28.7 | 48.6 | 51.5 | 46.7 | 37.7±0.05 | +6.5 |
| | | | | | | | | *ViT-B/16* | | | | | | | | | |
| NoAdapt | 29.1 | 29.6 | 31.6 | 31.2 | 25.1 | 39.3 | 31.5 | 24.6 | 30.2 | 54.3 | 64.5 | 48.4 | 34.2 | 52.5 | 55.1 | 38.8±0.00 | - |
| EM (Tent) | 49.4 | 54.2 | 56.3 | 46.9 | 48.1 | 56.7 | 52.0 | 55.7 | 61.0 | 68.2 | 77.2 | 64.5 | 52.8 | 68.1 | 67.5 | 58.6±0.09 | +0.0 |
| CADF | 50.1 | 54.8 | 56.8 | 53.2 | 57.4 | 61.9 | 60.1 | 64.9 | 66.4 | 71.5 | 77.0 | 66.7 | 64.0 | 70.7 | 68.9 | 63.0±0.01 | +4.4 |
| DEM* | 50.8 | 57.1 | 59.3 | 56.4 | 58.7 | 62.6 | 62.0 | 65.9 | 66.2 | 72.0 | 76.6 | 66.6 | 66.5 | 71.4 | 69.1 | 64.1±0.05 | +5.5 |
| AdaDEM-Norm | 48.9 | 54.4 | 56.5 | 47.7 | 49.5 | 57.4 | 53.1 | 58.1 | 62.0 | 68.5 | 77.3 | 65.1 | 55.2 | 68.4 | 67.9 | 59.3±0.04 | +0.7 |
| AdaDEM-MEC | 54.4 | 58.0 | 58.7 | 52.7 | 55.5 | 60.6 | 57.2 | 62.4 | 64.1 | 70.1 | 77.7 | 65.2 | 59.9 | 70.7 | 68.8 | 62.4±0.14 | +3.8 |
| AdaDEM | 54.9 | 59.2 | 60.1 | 53.3 | 56.9 | 61.7 | 58.6 | 63.3 | 65.3 | 70.9 | 77.6 | 64.3 | 61.7 | 71.3 | 69.3 | 63.2±0.16 | +4.6 |

computational overhead caused by hyperparameter search and achieves the best performance on both single-domain and continual tasks.

### D.5 Sensitivity of AdaDEM to Optimizers and Learning Rates

Setting different learning rates for the optimizer leads to different learning efficiencies of the model on unlabeled samples. A larger learning rate corresponds to an aggressive optimization strategy, which can effectively promote the model's learning of limited online samples, improving the speed at which DNNs adapt to the target distributions. But it may also cause the model to easily overfit to a single data distribution, resulting in the catastrophic forgetting problem. A smaller learning rate adopts a more conservative optimization approach, accelerating the model's adaptation to the target distribution while ensuring that the performance of DNNs does not degrade significantly. As shown in Fig. 9, we compare the sensitivities of the classical EM, AdaDEM-Norm, AdaDEM-MEC, and AdaDEM to different learning rates. Existing TTA algorithms, especially Tent, which simply uses the classical EM to optimize DNNs, are susceptible to the settings of the optimizer and learning rate, making it necessary to carefully adjust the optimization strategy. However, we demonstrate that AdaDEM normalizes the contributions of samples with different certainties to model optimization and introduces the estimated prior label distribution as a regularizer. It alleviates the model's tendency to optimize low-confident samples and the overwhelming influence of dominate classes. Therefore, the sensitivity of AdaDEM to the learning rate is significantly lower than that of the classical EM.

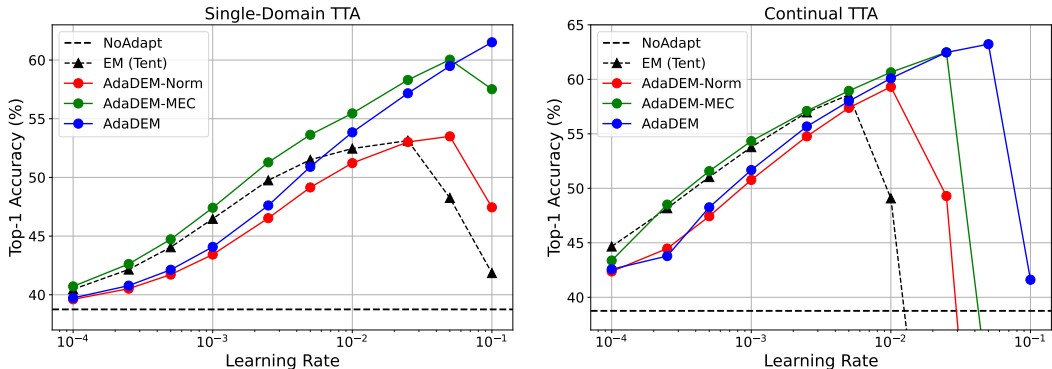

Figure 9: Comparison of the sensitivity to the learning rate among the classical EM, AdaDEM-Norm, AdaDEM-MEC, and AdaDEM for single-domain and continual TTA tasks.

Table 14: Ablation study on the role of $\delta$. We compare two methods for computing: $\delta$ derived from the reward brought from CADF, and $\delta_v$ derived from the reward brought from overall conditional entropy. AdaDEM and AdaDEM-V denote the approaches using $\delta$ and $\delta_v$ as reweighting factors.

| LR | Single-Domain TTA | | | LR | Continual TTA | | |
|---|---|---|---|---|---|---|---|
| | Classical EM | AdaDEM | AdaDEM-V | | Classical EM | AdaDEM | AdaDEM-V |
| 0.0001 | 40.5 | 39.7 | 39.9 | 0.0001 | 44.7 | 42.6 | **42.7** |
| 0.00025 | 42.2 | 40.8 | **41.0** | 0.00025 | 48.2 | 43.8 | 31.2 |
| 0.0005 | 44.1 | 42.1 | 39.4 | 0.0005 | 51.0 | 48.3 | 26.0 |
| 0.001 | 46.5 | 44.1 | 37.7 | 0.001 | 53.8 | 51.7 | 10.8 |
| 0.0025 | 49.8 | 47.6 | 29.2 | 0.0025 | 57.0 | 55.7 | 3.8 |
| 0.005 | 51.5 | 50.9 | 14.6 | 0.005 | **58.6** | 58.0 | 1.5 |
| 0.01 | 52.5 | 53.8 | 6.3 | 0.01 | 49.1 | 60.1 | 0.7 |
| 0.025 | **53.1** | 57.2 | 2.2 | 0.025 | 6.2 | 62.5 | 0.3 |
| 0.05 | 48.3 | 59.5 | 1.2 | 0.05 | 2.3 | **63.2** | 0.2 |
| 0.1 | 41.9 | **61.5** | 0.7 | 0.1 | 1.1 | 41.6 | 0.2 |

## D.6 Ablation Study on the Role of $\delta$ in AdaDEM

In Sec. 3.3, the parameter $\delta$ is introduced to reweight the conditional entropy $H(\mathbf{z})$ in AdaDEM. $\delta$ is defined as the L1-norm of the reward brought from CADF (*i.e.*, $\delta = \| - \partial T(\mathbf{z}|x,\theta)/\partial \mathbf{z}\|_1$), which quantifies the extent of changes in the model's output logits before and after EM optimization. Another common reweighting method uses the L1-norm of the reward from the overall loss function as the reweighting factor, specifically employing the partial derivative of the conditional entropy $H(\mathbf{z})$ with respect to the logits $\mathbf{z}$, *i.e.*, $\delta_v = \| - \partial H(\mathbf{z}|x,\theta)/\partial \mathbf{z}\|_1$. We compare these two methods for constructing AdaDEM using $\delta$ and $\delta_v$, respectively. Experimental results in Table 14 demonstrate that employing $\delta_v$ leads to degraded performance and robustness, whereas $\delta$ must be computed from CADF gradients. Since $\delta_v$ incorporates gradients from the overall loss function, it risks introducing penalties from components such as GMC or MEC into the normalization process. This may undermine the targeted mitigation of reward collapse by diluting CADF-specific rewards and altering optimization dynamics, such as misaligning sample contributions or impairing robustness in noisy or dynamic environments.

## D.7 Impact of Adding $\alpha$ to Scale MEC in AdaDEM

Similar to introducing $\alpha$ in the GMC of DEM to scale the penalty on logits as mentioned in Sec. 3.2, we can also introduce an $\alpha$ in AdaDEM to scale the effect of MEC. We conduct an ablation study on adding $\alpha$ to AdaDEM for single and continual TTA tasks, as shown in Fig. 10. The results indicate that AdaDEM-Norm is less sensitive to the value of $\alpha$ than AdaDEM-MEC. Meanwhile, changing the value of $\alpha$ also improves the performance of AdaDEM, but additional computational overhead is required to determine an appropriate value for $\alpha$.

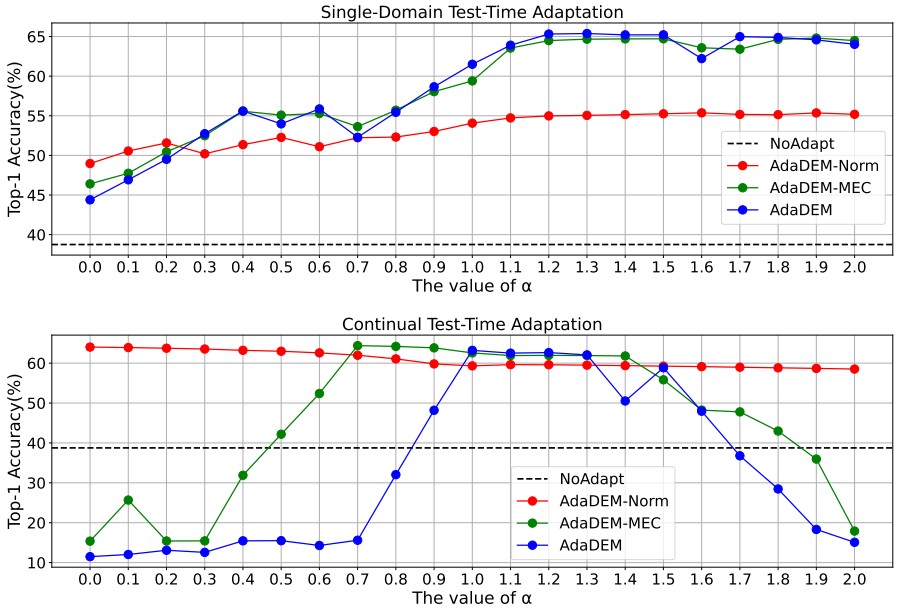

Figure 10: Ablation study of adding different $\alpha$ to scale MEC in AdaDEM.

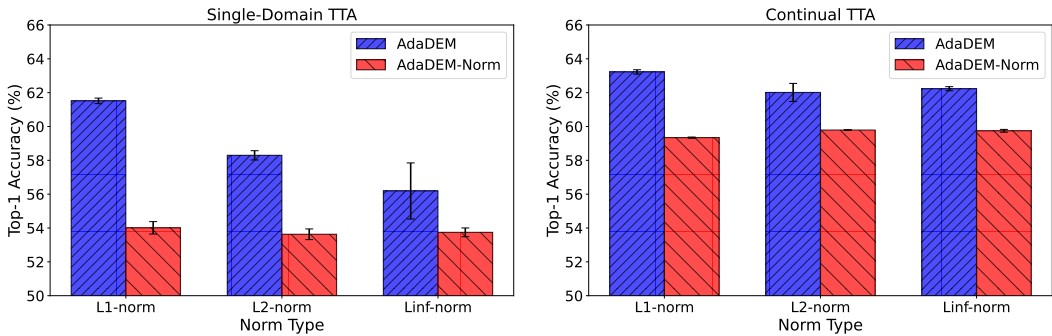

Figure 11: Ablation study of different Ln-norm types to normalize the reward brought from CADF.

## D.8 Impact of Different Norm Types on AdaDEM

As stated in Sec. 3.3, we default to using the L1-norm of the reward brought from CADF, *i.e.*, $\delta = \|\partial T/\partial \mathbf{z}\|_1$, to normalize the loss of an individual sample. In Fig. 11, we compare applying L1-norm, L2-norm, and L$\infty$-norm as criteria for measuring the reward of CADF on logits, which are then used to normalize the loss function. The results show that the impact of different norm types on AdaDEM-Norm is negligible. However, the L1-norm achieves better performance for AdaDEM compared to the other two methods. The reason is that the L1-norm directly measures the Manhattan distance of the logits predicted by the model for a sample before and after EM optimization.

## D.9 Impact of Different $\pi$ on AdaDEM

As stated in Sec. 3.3, we use the exponential moving average of the marginal entropy predicted by the model for each class as the estimated prior label distribution. Further, we can change the momentum $\pi$ in the moving average equation, *i.e.*, $\overline{\mathfrak{p}}_k^t = (1 - \pi) \cdot \overline{\mathfrak{p}}_k^{t-1} + \pi \cdot \mathfrak{p}_k$, to alter the follow-up ability of the estimated label distribution to the model's prediction probability. Specifically, we set $\pi = 1.0, 0.1$, and $0.01$ for the ablation study, as shown in Fig. 12. When $\pi = 1.0$, MEC uses the marginal entropy estimated for the samples in the current batch as the regularizer. When $\pi$ approaches $0.0$, the update of the estimated label distribution becomes slow. The experimental results show that in the single-domain task, $\pi = 0.1$ is suitable for AdaDEM and AdaDEM-MEC. However, for

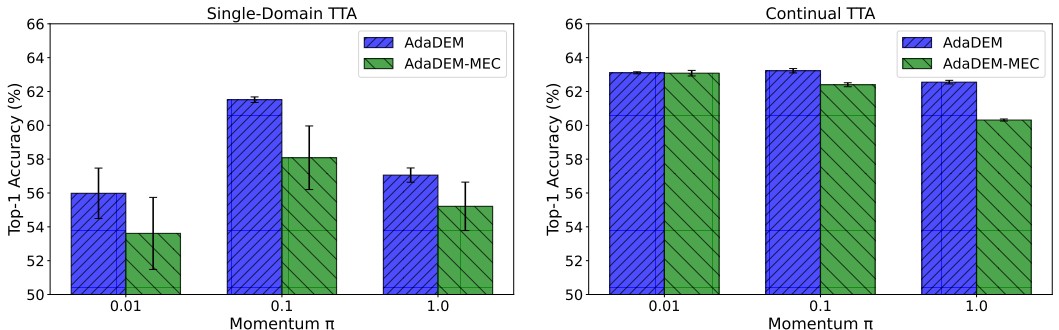

Figure 12: Ablation study of different $\pi$ to estimate prior label distribution for MEC.

Table 15: Imbalance study on Classical EM and AdaDEM. Top-1 classification accuracy (%) is reported. $\rho$ denotes the sample ratio between the most and least populous classes. $\Delta$ represents the accuracy improvement of AdaDEM over Classical EM.

| Methods | CIFAR-10-LT | | | Methods | CIFAR-100-LT | | |
|---|---|---|---|---|---|---|---|
| | $\rho = 100$ | $\rho = 500$ | $\rho = 1000$ | | $\rho = 10$ | $\rho = 100$ | $\rho = 500$ |
| Classical EM | 85.4 | 83.9 | 82.1 | Classical EM | 60.0 | 58.6 | 57.6 |
| AdaDEM | 91.4 | 90.7 | 88.6 | AdaDEM | 66.2 | 64.4 | 64.1 |
| $\Delta$ | +6.0 | +6.8 | +6.5 | $\Delta$ | +6.2 | +5.8 | +6.5 |
| Methods | EuroSat-LT | | | Methods | TissueMNIST-LT | | |
| | $\rho = 100$ | $\rho = 500$ | $\rho = 1000$ | | $\rho = 100$ | $\rho = 500$ | $\rho = 1000$ |
| Classical EM | 67.1 | 65.7 | 63.4 | Classical EM | 47.8 | 47.4 | 46.8 |
| AdaDEM | 76.6 | 74.8 | 73.6 | AdaDEM | 49.6 | 48.7 | 48.4 |
| $\Delta$ | +9.5 | +9.1 | +10.2 | $\Delta$ | +1.8 | +1.3 | +1.6 |

continual tasks, a smaller value of $\pi$ improves the performance of the algorithms for both AdaDEM and AdaDEM-MEC.

### D.10 Ablation Study on Imbalanced Benchmarks

Table 15 presents experimental results on CIFAR-10-LT, CIFAR-100-LT, EuroSat-LT, and TissueMNIST-LT. In addition to CIFAR-LT, we employ EuroSat-LT (focusing on land use and land cover classification) and TissueMNIST-LT (targeting biomedical scenarios and tissue type classification). We vary the configuration of $\rho$ (the sample ratio between the most and least populous classes). A larger $\rho$ indicates greater skewness in class sample sizes and higher imbalance severity. Results in Table 15, along with Fig. 5 in the manuscript, demonstrate that AdaDEM more effectively mitigates easy-class bias and further improves accuracy compared to the classical EM across tasks with varying degrees of imbalance.

## E  Broader Impacts

This paper aims to improve the widely-used classical Entropy Minimization method in the machine learning community. Our proposed method achieves credible performance across multiple machine learning tasks, including semi-supervised and unsupervised learning, domain adaptation, and reinforcement learning, thereby directly facilitating real-world applications of self-supervised entropy minimization strategies. Our work involves no human subjects and complies with legal requirements, with no anticipated harmful consequences. Through this study, we seek to enhance research and societal awareness regarding imperfectly supervised machine learning and deep learning systems.

