# OpenReview forum: "Decoupled Entropy Minimization"
_NeurIPS.cc/2025/Conference — NeurIPS 2025 poster_

### Official Review · Reviewer_xd1y · 2025-06-12

**Clarity:** 2
**Significance:** 2
**Originality:** 2
**Rating:** 4
**Confidence:** 4

**Summary:**

In this paper, the authors propose decomposing the entropy minimization objective into two distinct components, each responsible for a different task, one pushing for a peaked output distribution and one penalizing high-confidence classes. Next, they introduce a temperature factor to control reward collapse for highly certain samples, and a separate weighting factor to control the regularization effect of the second component. To avoid the need for manual hyperparameter search, they propose an adaptive variant that automatically adjusts the weight of the loss based on the norm of gradients. The authors conduct extensive experiments on a wide range of datasets, demonstrating improved performance.

**Questions:**

- Could the authors clarify the exact connection between DEM and AdaDEM? Specifically, is AdaDEM intended to be a direct reformulation of DEM with adaptive scaling, or does it represent a fundamentally different objective? How the hyper-pararameters are related to the gradient-based rescaling of the objective?

- Given that $\delta$ δ is computed as the norm of the gradients of $T$, have the authors explored or could provide discussion on the effect of using the norm of the full loss gradients instead? Would this lead to different behavior, and if so, how?

- Eq. (8) appears to be very close to standard conditional entropy with a sample-based normalization term. Could the authors clarify how AdaDEM differs in practice from standard conditional entropy minimization under this view? What about comparisons with existing sample based reweighting approaches? Should the method be compared against them given the high similarity with such approaches?

-  Regarding the term introduced in lines 161–162 and its role in the derivation of Eq. 8, could the authors provide further justification for this simplification?  Why is this particular simplification needed or desirable?

**Ethical Concerns:**

["NO or VERY MINOR ethics concerns only"]

**Final Justification:**

After considering the authors’ rebuttal, I am increasing my rating to borderline accept. However, I still have concerns regarding the incremental contribution and the novelty over existing formulations.

**Limitations:**

Yes.

**Paper Formatting Concerns:**

No.

**Quality:**

3

**Strengths And Weaknesses:**

The paper is overall well written and is supplemented with extensive experiments that demonstrate the effectiveness of the proposed approach. The idea of the proposed decomposition of entropy is interesting and can provide an alternative view on these entropy objectives, which is positive. However, the ideas presented in the paper are loosely connected and not well motivated. Please find some important comments below:

- The connection between DEM and AdaDEM is somewhat vague and not very clear. If my understanding is correct, AdaDEM does not use the temperature factor or the $\alpha$ parameter. So, AdaDEM does not remove these hyperparameters by providing adaptive estimations for them, but rather by using a different formulation. In fact, AdaDEM is Eq. 8 with the added  $\frac{1}{\delta}$  factor. $\delta$ is calculated as the norm of gradients of T. Therefore, even though AdaDEM is presented as an attempt to remove the hyperparameters of DEM, it seems to me that AdaDEM can be directly derived from DEM without introducing any hyperparameters. This is also supported by the signficantly better performance of AdaDEM compared to DEM.

- This also brings out another quite important concern. If (8) is equivalent to (4), then AdaDEM is actually the regular conditional entropy with just a sample-based normalization term. Another question that naturally arises here is: what if we do not use the gradients of  $T$ to calculate $\delta$, but rather the whole loss? Appropriate supporting discussion and/or ablation studies are needed here to support this. Furthermore, this approach is very similar to prior works on reweighting samples based on gradient directions, e.g., Ren, Mengye, et al. "Learning to reweight examples for robust deep learning." International Conference on Machine Learning. 2018, as well as gradient norm approaches, e.g., Chen, Zhao, et al. "GradNorm: Gradient normalization for adaptive loss balancing in deep multitask networks." International Conference on Machine Learning. PMLR, 2018, and similar extensions.

- The motivation for introducing the term proposed in lines 161–162 should be better justified by the authors. It is not directly evident how  p_i  should be computed or why we need to perform this simplification that leads to Eq. 8. This is a minor concern.

Overall, the paper provides strong performance on benchmark datasets and extensive comparisons, demonstrating the effectiveness of the proposed approach. However, I find that there are significant gaps in the presentation and motivation of the proposed method, so I am reluctant to provide an acceptance decision at this stage. I appreciate the insights provided by the discussion on entropy decomposition, but it seems that the proposed method essentially amounts to a relatively simple sample-based reweighting scheme, a concept that has already been explored in various forms in prior work.

---

> ### Author Rebuttal · Authors · 2025-07-31
>
> We sincerely appreciate the time and effort put into reviewing our paper and providing valuable feedback. We would like to address your questions below.
>
> > **R4Q1**: Questions about the connection between DEM and AdaDEM, specifically how AdaDEM eliminates hyperparameters without adaptive estimation and whether it is directly derivable from DEM.
>
> AdaDEM is derived from the decoupled framework of classical EM (as in DEM) but fundamentally differs by replacing the hyperparameter-dependent components: it normalizes the Cluster Aggregation Driving Factor (CADF) reward using the L1-norm of its gradient ($δ = || -\partial T / \partial z ||\_1$) and substitutes the Gradient Mitigation Calibrator (GMC) with a Marginal Entropy Calibrator (MEC) for dynamic prior estimation. This reformulation, expressed in Eq. 12, avoids $\tau$ and $\alpha$ by design, not through adaptive hyperparameter tuning, and directly addresses DEM's limitations (e.g., reward collapse and easy-class bias), leading to superior performance without introducing new hyperparameters.
>
> > **R4Q2**: Whether AdaDEM is equivalent to normalized conditional entropy and requests clarification on using gradients of the entire loss versus CADF for $\delta$.
>
> AdaDEM is not equivalent to simply normalizing classical entropy. Eq. 8 reformulates entropy via decoupled components (CADF/GMC), but AdaDEM replaces GMC with MEC and normalizes using $\delta$ derived exclusively from CADF’s gradients (Eq. 12). Ablations in Table A validate that $\delta$ must be computed from CADF gradients—using the whole loss degrades performance. Such a substitution of using the full loss would likely introduce penalties from components like GMC or MEC into the normalization, potentially disrupting the targeted mitigation of reward collapse by diluting CADF-specific rewards and altering optimization dynamics, such as misaligning sample contributions or affecting robustness in noisy/dynamic environments.
>
> Table A: Ablation study of $\delta$. AdaDEM-V means that using $\delta$ computed from the whole loss.
>
> | Single-Domain TTA |      |        |          | Continual TTA |      |        |          |
> | ----------------- | ---- | ------ | -------- | ------------- | ---- | ------ | -------- |
> | LR                | EM   | AdaDEM | AdaDEM-V | LR            | EM   | AdaDEM | AdaDEM-V |
> | 0.0001            | 40.5 | 39.7   | 39.9     | 0.0001        | 44.7 | 42.6   | **42.7**     |
> | 0.00025           | 42.2 | 40.8   | **41.0**     | 0.00025       | 48.2 | 43.8   | 31.2     |
> | 0.0005            | 44.1 | 42.1   | 39.4     | 0.0005        | 51.0 | 48.3   | 26.0     |
> | 0.001             | 46.5 | 44.1   | 37.7     | 0.001         | 53.8 | 51.7   | 10.8     |
> | 0.0025            | 49.8 | 47.6   | 29.2     | 0.0025        | 57.0 | 55.7   | 3.8      |
> | 0.005             | 51.5 | 50.9   | 14.6     | 0.005         | **58.6** | 58.0   | 1.5      |
> | 0.01              | 52.5 | 53.8   | 6.3      | 0.01          | 49.1 | 60.1   | 0.7      |
> | 0.025             | **53.1** | 57.2   | 2.2      | 0.025         | 6.2  | 62.5   | 0.3      |
> | 0.05              | 48.3 | 59.5   | 1.2      | 0.05          | 2.3  | **63.2**   | 0.2      |
> | 0.1               | 41.9 | **61.5**   | 0.7      | 0.1           | 1.1  | 41.6   | 0.2      |
>
> > **R4Q3**: Discussion of relations to prior gradient-based reweighting methods.
>
> The design of AdaDEM addresses EM-specific limitations (reward collapse/easy-class bias), distinct from general sample reweighting schemes. Unlike supervised reweighting (Ren et al.) or multitask balancing (GradNorm), AdaDEM’s $\delta$ is unsupervised, dynamically adjusted per-sample via the L1-norm of CADF’s reward, and integrated with MEC for distribution calibration without label priors. The approach is tailored for EM’s decoupled dynamics, as evidenced by gains over baselines (Tables 1–4).
>
> > **R4Q4**: Why introducing the term in lines 161–162 and clarifying the motivation for the simplification leading to the $\mathbf{p}_i$ in Eq. 8.
>
> We clarify that $\mathbf{p}_i$ is introduced as a constant (excluded from the computation graph) to reformulate the conditional entropy equivalently to Eq. 6–7, enabling identical partial derivatives as GMC while simplifying gradient computation by avoiding dependencies on variable probabilities. This rewrite (Eq. 8) facilitates practical implementation and analysis of decoupled effects without altering the optimization objective, as $\mathbf{p}_i$ requires no explicit computation beyond its role as a fixed scaling term.

---

> ### Comment · Reviewer_xd1y · 2025-08-02
>
> Thank you for your responses. The additional experiments provided have clarified one of my main concerns.  Therefore, I am increasing my score.

---

> > ### Author Response · Authors · 2025-08-09
> >
> > We are delighted to learn that our explanations addressed your concerns and appreciate the increased score. Thank you again for your expertise and invaluable feedback in enhancing the quality of our paper!
> >
> > Best Regards

---

### Official Review · Reviewer_pbbx · 2025-06-19

**Clarity:** 1
**Significance:** 2
**Originality:** 3
**Rating:** 4
**Confidence:** 5

**Summary:**

This paper explores the reformulation and decouple of classical entropy minimisation into two parts.

One part is namely as cluster aggregation driving factor rewards and another one is called gradient mitigation calibrator.

They identify the weaknesses of classical EM, and propose adaptive decoupled entropy minimisation.

The evaluations and analysis are performed on several benchmarks, covering different learning objectives, and show consistent improvements compared to the use of classical EM.

**Questions:**

Please refer to weaknesses for questions.

**Ethical Concerns:**

["NO or VERY MINOR ethics concerns only"]

**Final Justification:**

The authors have addressed the concerns I raised in a clear and thorough manner.

I have also carefully reviewed the comments from the other reviewers, along with the authors’ responses.

I believe this work presents some meaningful insights. However, I also acknowledge and share some of the reservations expressed by reviewers who maintain a borderline assessment.

That said, based on the improvements in the revised version, I am satisfied with the authors’ efforts and am happy to raise my score from a borderline reject to a borderline accept, aligning with the direction suggested by other reviewers (Borderline).

**Limitations:**

Limitations have been discussed in conclusion section.

**Quality:**

3

**Strengths And Weaknesses:**

strengths:

+ The idea of decouple classical EM into two parts where each part serves a different aim is interesting and new.

+ The comprehensive experiments, analysis, evaluations and discussions somehow show the effectiveness of the proposed model.

+ The paper is overall well written, somehow clean and clear.

weaknesses:

Major:

- In figure 1, some terms are not properly explained and unclear to reviewer. For example, the introduction of $\alpha$, $\tau$, $\delta$, as well as some unknown operations appear to be too early. This figure does not present much insights or novel elements for readers. In addition, as the first figure in the paper, it does not seem to be attractive to readers at all. Also is it necessary to have 3 unclear equations and terms presented in the figure, and are there better ways to present them in a clear, interesting, and easy-to-understand manner?

- This paper appears to have several redundancies. For example, Line 107-108 is the same as Line 19-20; Line 114-116 is the same as Line 23-24. These sentences look almost identical and just the references differ. This is inappropriate.

- It would be better to be clear eg what are scalars, vectors, and matrices. Some equations appear to be a bit strange. For example, eq 4, left side input is $z$, but the right side there is no $z$, although $p$ and $z$ have connections, as a research paper, it would be better to make it clear and rigorous, rather than a technical report with some levels of inconsistencies.

- Fig 3 and 4 right subfigures, what are the optimal values used? the right most two subfigures are not being properly explained. For example, what are those numbers mean inside brackets, shouldn't the readers need to know those concepts? Regarding NoAdapt, shouldn't the plot (yellow lines) the same in both plots? These concepts are not being properly explained and made clearer enough to readers.

- Line 177-180 is unclear, for example, this part refers to figure 3 left and eq 7, but eq 7 does not contain either $\alpha$ and $\tau$ at all. the motivation for $\alpha$ and $\tau$ is unclear. What motivate the use of $\alpha$ and $\tau$?

- Some of the operations are not being clearly explained, for example, Line 216, and eq 12 regarding $-t$ and its roles are unclear.

- Line 248, "on multiple random seeds", how many runs exactly? The reviewer thinks that as a research paper, it is necessary to be as clear as possible, also what are the random seeds, are random seeds in random manner when calculating mean and std?

- Line 296-299, table 1 and 2, especially 2, seem does not reflect on the discussions made in this part?

Minor:

- Line 142, what is that $\mathcal{L}$, this is not being properly introduced / explained before using it. Also in Line 142-144, it is unclear why such operation is performed. It would be better to provide justifications.

- apart from redundancies in texts, figures also have some levels of redundancies. For example, fig 3 left, and figure 4 left, why $\alpha=1.0$ appears 3 times, and $\tau=1.0$ appears 3 times in both legends.

- Using `\eqref` for referencing equations, instead of using `\ref`.

- table 1, DEM* should be explained in caption.

- The reviewer also reads the appendix. Page 22, figure 7 top row, texts and figures a bit cluttered.

---

> ### Author Rebuttal · Authors · 2025-07-31
>
> We sincerely appreciate the time and effort put into reviewing our paper and providing valuable feedback. We would like to address your questions below.
>
> > **R3Q1**: Concerns about Fig. 1.
>
> We revise Fig. 1 to explicitly label $\alpha$, $\tau$, and $\delta$ using annotations. Core novelty is emphasized with intuitive visual cues. The figure is reorganized as a conceptual flowchart comparing classical EM, DEM*, and AdaDEM, using succinct text and highlighting information flow.
>
> > **R3Q2**: Concerns about redundancies in the manuscript.
>
> These repetitions are fully eliminated in the revised version to ensure conciseness and adherence to publication standards. Thank you for highlighting this issue.
>
> > **R3Q3**: Concerns about clearer definitions in equations.
>
> We thank the reviewer for the constructive feedback on enhancing mathematical rigor. We agree that explicitly denoting $p$ as a function of $z$ would improve clarity, and we revise all equations to rigorously define scalars, vectors, and matrices, ensuring consistency throughout the paper.
>
> > **R3Q4**: Questions about Fig. 3 and Fig. 4.
>
> We acknowledge the need for clearer explanations. In Fig. 3 (right), the optimal values of $\tau$ are represent in Fig. 3 (center), and the optimal values of $\alpha$ in Fig. 4 (right) are $\alpha=0.0$ for single-domain TTA and $\alpha=1.3$ for Continual TTA. NoAdapt (yellow) represents the consistent source-model performance without adaptation, so that the results are the same for both single-domain and continual TTA tasks. We clarify brackets as "mean±std" and add captions to define NoAdapt uniformly.
>
> > **R3Q5**: Questions about the motivation for introducing $\alpha$ and $\tau$.
>
> The reference to Eq. 7 in lines 177-180 serves to contextualize the gradient behavior before introducing $\alpha$ and $\tau$. $\tau$ in Eq. 9 reshapes the reward curve in CADF to enhance contributions from high-confidence samples (Fig. 3 left), while $\alpha$ in Eq. 10 scales GMC penalties to prevent overfitting in noisy/dynamic environments (Fig. 4 center), both optimizing robustness and performance upper bounds in DEM*.
>
> > **R3Q6**: Concerns about Eq. 12.
>
> In Eq. 12 and the surrounding text (e.g., line 216), $t$ denotes the iteration index for the exponential moving average in the Marginal Entropy Calibrator (MEC), where $\overline{\mathbf{p}}\_{k}^t$ represents the estimated probability vector for class $k$ at step $t$, dynamically updated as $\overline{\mathbf{p}}\_{k}^{t} = 0.9 \cdot \overline{\mathbf{p}}\_{k}^{t-1} + 0.1 \cdot \mathbf{p}\_{k}$ using the current prediction $\mathbf{p}\_{k}$, thereby replacing GMC to mitigate easy-class bias without label priors.
>
> > **R3Q7**: Concerns about random seeds.
>
> In the paper, we use 3 random seeds for all experiments, specifically $\{1, 2, 3\}$. The random seeds are chosen to determine test sample order or initial conditions, ensuring reproducibility, and the mean and standard deviation are computed across these runs to account for variability, as specified in Sec. 4.1 and Appendix B.1.
>
> > **R3Q8**: Whether the discussions in lines 296-299 and Tables 1-2 are adequately reflected in the paper.
>
> The paper explicitly addresses the results in Table 1 and Table 2 within the main text: Table 1's ablation findings on CADF and robustness are discussed in Sec. 3.2, while Table 2's TTA performance gains with DEM*/AdaDEM are analyzed in Sec. 4.2. Lines 296-299 directly reference Table 2 to explain how sample selection methods like EATA/SAR are enhanced, mitigating reward collapse and easy-class bias, which aligns with the tables' data on accuracy improvements and $\Delta$ values). This ensures coherence between results and discussion.
>
> > **R3Q9**: What is the $\mathcal{L}$?
>
> $\mathcal{L}$ refers to the entropy minimization loss $H(z)$ (Eq. 5), which is the conditional entropy minimized in EM.
>
> > **R3Q10**: Concerns about the redundant labeling in Fig.3 and Fig. 4.
>
> We acknowledge the redundancy in the legends of Fig. 3 (left) and Fig. 4 (left), where $\alpha=1.0$ and $\tau=1.0$ are repeated for clarity in the original plot. This will be rectified in the final version by retaining only unique labels (e.g., one instance of $\alpha=1.0$ and $\tau=1.0$) to eliminate redundancy while preserving interpretability.
>
> > **R3Q11**: Suggestions of using \eqref to replace \ref, explaining DEM* in Table 1's caption, and address the cluttered presentation in Fig. 7.
>
> We acknowledge the reviewer's feedback and optimize the manuscript to enhance clarity by: (1) Using \eqref for equation references, (2) Adding an explanation of DEM* to Table 1's caption, and (3) Modifying Fig. 7's layout to improve visual spacing.

---

> > ### Comment · Reviewer_pbbx · 2025-08-02
> >
> > Thank you for the responses.
> >
> > Could the authors kindly provide the specific modifications and revisions that have been made, for example, **what updates** were added and **in which sections**?
> >
> > Sharing these details now would strengthen the submission and allow the reviewer to **more clearly assess the improvements**. The reviewer is particularly interested in understanding the **exact changes** implemented.

---

> > > ### Author Response · Authors · 2025-08-04
> > >
> > > Thank you for your follow-up query and constructive guidance. Due to current NeurIPS policy restrictions, we cannot submit a revised manuscript for review. Nevertheless, we are pleased to detail the specific revisions made to address your initial concerns. Below is a summary of key updates, cross-referenced with relevant sections/figures/tables.
> > >
> > > - **Fig.1 in Sec.1**: We have replaced original schematic with a conceptual flowchart comparing the classical EM, DEM*, and AdaDEM. We have added intuitive arrows for information flow and color-coding for components. We have explicitly labeled CADF, GMC, and MEC using annotations and described $\alpha$, $\tau$, and $\delta$ using succinct text. We have removed redundant equations and retained only Eq. 6 (decoupling) for clarify.
> > > - **Sec. 2 (Lines 107-108 and 114-116)**: We have deleted duplicated sentences in Sec. 2 (Lines 107-108 and 114-116), and have elaborated in greater detail on the application of Entropy Minimization within related works rather than simply citing them.
> > > - **Sec. 3 (All equations)**: We have revised equations to explicitly define $H(\mathbf{z})$ as a function of logits $\mathbf{z}$. We also have revised equations to rigorously define scalars, vectors, and matrices. For example, Eq. 4 has been revised to $H(\mathbf{z}) = - \sum_{i=1}^C p_i(\mathbf{z}) \log p_i(\mathbf{z})$.
> > > - **Table 1 in Sec. 3**: We have explained DEM* in the caption as "DEM* searches the optimal hyperparameters $(\tau*, \alpha*)$ on a subset of target data."
> > > - **Sec. 3.1 (Line 142)**: We have explicitly explained that the loss function $\mathcal{L}=H(\mathbf{z})$.
> > > - **Fig. 3 \& 4 in Sec. 3.2**: We have added caption: "Optimal $\tau$ sourced from Fig. 3 (center)." and "Optimal $\alpha=1.0$ for single TTA and optimal $\alpha=1.3$ for continual TTA." We have unified explanation of NoAdapt as baseline source-model performance without adaptation. We have removed redundant $\alpha$ and $\tau$ labels and retained only unique values (i.e., one instance of $\alpha=1.0$ and $\tau=1.0$).
> > > - **Sec. 4.1 (Line 248)**: We have explicitly stated "All experimental results are averaged over 3 fixed seeds $(1,2,3)$ controlling test-sample order and initial conditions."
> > > - **Sec. 4.3 (Line 296-299)**: We have added direct cross-references: "Table 2 shows the gains of applying AdaDEM to different optimizers like vanilla SGD (Tent$\dag$), Momentum (ETA, EATA, DeYO), Adam (Tent), and SAM (SAR)."
> > > - **Fig. 7 in Appendix**: We have updated Fig. 7 and increased the spacing between text and figures to enhance visual clarity.

---

> > > > ### Comment · Reviewer_pbbx · 2025-08-05
> > > > **Unfortunitely my concerns are not being carefully addressed**
> > > >
> > > > The updates provided are still lacking in sufficient detail.
> > > >
> > > > The reviewer would like to see exactly what modifications have been made in response to the comments. For instance, when addressing feedback such as:
> > > >
> > > > "This paper appears to have several redundancies. For example, Line 107–108 is the same as Line 19–20; Line 114–116 is the same as Line 23–24. These sentences look almost identical and only the references differ. This is inappropriate."
> > > >
> > > > it would be much more effective and helpful to state explicitly, for example:
> > > >
> > > > “We have revised Paragraph 3 as follows:
> > > > [Revised paragraph text]”
> > > >
> > > > Similarly, in response to:
> > > >
> > > > "Line 177–180 is unclear; for example, this part refers to Figure 3 (left) and Eq. 7, but Eq. 7 does not contain...?"
> > > >
> > > > please specify exactly which section and lines have been updated and how, such as:
> > > >
> > > > “We have revised Section X, Lines Y–Z, as follows:
> > > > [Updated text]”
> > > >
> > > > Stating only that revisions “have been made” or “will be made” may give the impression that the changes are incomplete, potentially leading the reviewer to anticipate another round of revision. This could negatively impact the evaluation of the manuscript.
> > > >
> > > > Could the authors kindly address each concern with care and clearly present the specific revisions made in response? High-quality work requires effort, and demonstrating this effort **now** will strengthen the manuscript and support a more favorable review outcome.

---

> ### Author Response · Authors · 2025-08-06
>
> Thank you for your constructive guidance. In response to the comments, we provide detailed revisions below. We sincerely hope our clarifications are helpful in improving your opinion of our work.
>
> > **Q1**: In figure 1, some terms are not properly explained and unclear to reviewer. For example, the introduction of $\alpha$, $\tau$, $\delta$, as well as some unknown operations appear to be too early. This figure does not present much insights or novel elements for readers. In addition, as the first figure in the paper, it does not seem to be attractive to readers at all. Also is it necessary to have 3 unclear equations and terms presented in the figure, and are there better ways to present them in a clear, interesting, and easy-to-understand manner?
>
> We have updated **Figure 1**, replacing the original schematic with a conceptual flowchart comparing classical EM, DEM\*, and AdaDEM. Intuitive arrows are added to illustrate information flow from the model's output logits to the computed entropy (loss), while color-coding highlights methodological differences among classical EM, DEM\*, and AdaDEM. Parameters $\tau$, $\alpha$, and $\delta$ are explicitly labeled and described with the succinct annotations: "temperature parameter to soften probabilities", "weight factor to scale GMC", and "L1-norm of the gradient from CADF". For clarity, redundant equations are removed, retaining only Eq. (6) (decoupling).
>
> > **Q2**: This paper appears to have several redundancies. For example, Line 107-108 is the same as Line 19-20; Line 114-116 is the same as Line 23-24. These sentences look almost identical and just the references differ. This is inappropriate.
>
> We have revised **Section 2, Lines 107-108**, as follows:
>
> EM has been demonstrated effective for clustering (avoiding trivial solutions where most instances concentrate in a single cluster) [2;11;19], semi-supervised learning (enhancing model prediction accuracy per data point) [1;3;9], and unsupervised learning (yielding peaked conditional class distributions) [15;19].
>
> We have revised **Section 2, Lines 114-116**, as follows:
>
> EM is widely used in active learning (reducing the number of possible hypotheses as rapidly as possible) [6;14], domain adaptation (aligning the target data distribution with the source data distribution) [7;19], and online learning (connecting entropy to error and shift) [5;8;10;20;21].
>
> > **Q3**: It would be better to be clear eg what are scalars, vectors, and matrices. Some equations appear to be a bit strange. For example, eq 4, left side input is $z$, but the right side there is no $z$, although $p$ and $z$ have connections, as a research paper, it would be better to make it clear and rigorous, rather than a technical report with some levels of inconsistencies.
>
> We have revised **Equation 4** as: $H(\mathbf{z}) = - \sum_{i=1}^C p_i(\mathbf{z}) \log p_i(\mathbf{z})$.
>
> We have revised **Equation 5** as: $\Theta^* = \arg\min_{\Theta} - \sum_{\mathbf{x}}^X \sum_{i=1}^C p_i(\mathbf{x}, \Theta) \log p_i(\mathbf{x}, \Theta)$.
>
> We have revised **Equation 6** as: $H(\mathbf{z}) = - \sum_{i=1}^C p_i(\mathbf{z}) \log \frac{e^{z_i}}{\sum_{j=1}^C e^{z_j}} = - \sum_{i=1}^C p_i(\mathbf{z}) z_i + \log \sum_{i=1}^C e^{z_i}$.
>
> We have revised **Equation 8** as: $H(\mathbf{z}) = - \sum_{i=1}^C (p_i(\mathbf{z}) - \mathfrak{p}_i)z_i$.
>
> We have revised **Equation 11** as: $H(\mathbf{z}) = T_\tau(\mathbf{z}) + Q_\alpha(\mathbf{z}) = - \sum_{i=1}^C p_{\tau i}(\mathbf{z}) z_i + \alpha \log \sum_{i=1}^C e^{z_i}$.
>
> We have revised **Equation 12** as: $H(\mathbf{z}) = - \frac{1}{\delta} \sum_{i=1}^C (p_i(\mathbf{z}) - \overline{\mathfrak{p}}_{k i}^t) z_i$.
>
> > **Q4**: Fig 3 and 4 right subfigures, what are the optimal values used? the right most two subfigures are not being properly explained. For example, what are those numbers mean inside brackets, shouldn't the readers need to know those concepts? Regarding NoAdapt, shouldn't the plot (yellow lines) the same in both plots? These concepts are not being properly explained and made clearer enough to readers.
>
> We have revised **the caption of Figure 3 (Right)** as follows:
>
> (**Right**) Detailed TTA results using the optimal $\tau$ across $15$ target domains, with exact values sourced from Fig. 3 (Center). "NoAdapt" denotes the baseline using fixed source model parameters without adaptation, hence its performance remains consistent for both single-domain and continual TTA tasks. Values in brackets indicate the corresponding axis range.
>
> We have revised **the caption of Figure 4 (Right)** as follows:
>
> (**Right**) Detailed TTA results across $15$ target domains, using the optimal $\alpha=1.0$ for single-domain TTA tasks and $\alpha=1.3$ for continual TTA tasks. These $\alpha$ values are selected based on the ablation results in Fig. 4 (Center). The definitions of "NoAdapt" and values in brackets are consistent with Fig. 3 (Right).

---

> > ### Comment · Reviewer_pbbx · 2025-08-07
> > **Q2-4 clear Q1 unclear**
> >
> > Thank you for the responses.
> >
> > As a rigorous reviewer, I carefully evaluate each point raised.
> >
> > Could the authors clarify whether the caption of Figure 1 has been revised?
> >
> > Regarding the revised points (Q2-Q4), they are clear and well addressed, thank you for your efforts.

---

> ### Author Response · Authors · 2025-08-06
>
> > **Q5**: Line 177-180 is unclear, for example, this part refers to figure 3 left and eq 7, but eq 7 does not contain either $\alpha$ and $\tau$ at all. the motivation for $\alpha$ and $\tau$ is unclear. What motivate the use of $\alpha$ and $\tau$?
>
> We have revised **Section 3.2, Lines 177-180**, as follows:
>
> We introduce a temperature $\tau$ in CADF to soften the probability $p_i$, thereby reshaping the reward curve. This operation is defined in Eq. (9), with its partial derivative *w.r.t.* logit $z_i$ given in Eq. (17). Based on Eq. (17), Fig. 3 (left) plots EM's reward curve across different $\tau$ values, which degenerates to classical EM when $\tau=1.0$. Notably, the reward collapses to $0.0$ when the predicted probability approaches either $1.0$ or $1/C$. The introduction of $\tau$ ensures high-probability predictions remain within the high-reward interval, thus mitigating reward collapse.
>
> > **Q6**: Some of the operations are not being clearly explained, for example, Line 216, and eq 12 regarding $-t$ and its roles are unclear.
>
> We have revised **Section 3.3, Lines 215-217**, as follows:
>
> The MEC dynamically estimates the marginal entropy during learning via an exponential moving average of $\overline{\mathfrak{p}}_k^0 = 1/N_k \sum^{N_k} \mathfrak{p}_k$ where $\mathfrak{p}_k\in \mathbb{R}^C$ is a probability vector of the $k$-th class. The estimated probability is dynamically updated as $\overline{\mathfrak{p}}_k^t = 0.9 \cdot \overline{\mathfrak{p}}_k^{t-1} + 0.1 \cdot \mathfrak{p}_k$ where $t$ denotes the iteration index.
>
> > **Q7**: Line 248, "on multiple random seeds", how many runs exactly? The reviewer thinks that as a research paper, it is necessary to be as clear as possible, also what are the random seeds, are random seeds in random manner when calculating mean and std?
>
> We have revised **Section 4.1, Lines 247-248**, as follows:
>
> All experimental results are obtained from $3$ fixed seeds $(1, 2, 3)$ controlling training/testing sample order and initial conditions. Performance metrics are reported as "mean ± standard deviation".
>
> > **Q8**: Line 296-299, table 1 and 2, especially 2, seem does not reflect on the discussions made in this part?
>
> We have revised **Section 4.3, Lines 296-299**, as follows:
>
> **Sensitivity of AdaDEM to Optimizers and Learning Rates.** We compare the impacts of different optimizers and learning rates on classical EM and AdaDEM.  As shown in Table 1, AdaDEM effectively reduces EM's sensitivity to the learning rate. Meanwhile, Table 2 demonstrates performance gains when applying AdaDEM to various optimizers including vanilla SGD (Tent$^{\dagger}$), Momentum (ETA, EATA, DeYO), Adam (Tent), and SAM (SAR). The results indicate that AdaDEM maintains compatibility with SGD or optimizers utilizing first-/second-order momentum estimation.
>
> > **Q9**: Line 142, what is that $\mathcal{L}$, this is not being properly introduced / explained before using it. Also in Line 142-144, it is unclear why such operation is performed. It would be better to provide justifications.
>
> We have revised **Section 3.1, Lines 142-144**, as follows:
>
> To analyze the contributions of the two decoupled parts with opposite effects from classical EM to the model's output, we define the terms "reward" and "penalty."  We define "reward" as the negative partial derivative of the loss function $\mathcal{L}=H(\mathbf{z})$ with respect to the logit $z_i$, *i.e.*, $-\partial \mathcal{L}/\partial z_i$, which aligns with the gradient descent direction for solving Eq. (5). We define "penalty" as the opposite of "reward", *i.e.*, $\partial \mathcal{L} / \partial z_i$. We subsequently demonstrate that minimizing CADF rewards the model's output logits, while minimizing GMC penalizes them.
>
> > **Q10**: Apart from redundancies in texts, figures also have some levels of redundancies. For example, fig 3 left, and figure 4 left, why $\alpha=1.0$ appears 3 times, and $\tau=1.0$ appears 3 times in both legends.
>
> We have revised **Figure 3 (Left) and Figure 4 (Left)**, removing redundant $\alpha=1.0$ labels in Figure 3 (Left) and $\tau=1.0$ labels in Figure 4 (Left), while retaining only one instance of $\alpha=1.0$ and $\tau=1.0$.
>
> > **Q11**: Using `\eqref` for referencing equations, instead of using `\ref`.
>
> We have replaced all `\ref` commands with `\eqref` for equation references.
>
> > **Q12**: Table 1, DEM* should be explained in caption.
>
> We have revised **the caption of Table 1 (Left)** as follows:
>
> (**Left**) Ablation studies in single-domain and continual TTA tasks. DEM* searches optimal hyperparameters $(\tau*, \alpha*)$ on a subset of target data. $\Delta$ denotes the performance improvement relative to the baseline.
>
> > **Q13**: The reviewer also reads the appendix. Page 22, figure 7 top row, texts and figures a bit cluttered.
>
> We have updated **Figure 7** and increased the spacing between text and figures to enhance visual clarity.

---

> > ### Comment · Reviewer_pbbx · 2025-08-07
> > **The remaining concerns (Q5–Q13) have been clearly addressed in this revision.**
> >
> > Thank you for the detailed responses and the effort put into the revision.
> >
> > The authors have addressed the points in a clear and thorough manner.
> >
> > Based on the revised version, I am satisfied with the improvements and happy to raise my score accordingly.

---

> > > ### Author Response · Authors · 2025-08-08
> > >
> > > We are pleased that our explanations helped improve your perspective on our work and appreciate your increased score.
> > >
> > > We have revised **the caption of Figure 1** as follows:
> > >
> > > Illustration of classical EM, DEM\*, and AdaDEM. EM is decoupled into two parts with opposite effects: CADF and GMC. DEM\* softens model's prediction via temperature $\tau$ and scales GMC via weight $\alpha$, searching for optimal $(\tau*, \alpha*)$ to maximize classical EM's performance. AdaDEM normalizes CADF reward by $\delta$ (L1-norm of the gradient) to prevent reward collapse, and replaces GMC with Marginal Entropy Calibrator (MEC) to reduce easy-class bias.
> > >
> > > Thank you again for your expertise and invaluable feedback in enhancing the quality of our paper!
> > >
> > > Best Regards

---

### Official Review · Reviewer_LD3y · 2025-06-28

**Clarity:** 3
**Significance:** 3
**Originality:** 3
**Rating:** 4
**Confidence:** 4

**Summary:**

The authors revisit classical entropy minimization (EM) and show that its conditional-entropy objective can be rewritten as two antagonistic terms: a cluster aggregation driving factor (CADF), which amplifies high-confidence predictions, and a gradient mitigation calibrator (GMC), which penalizes them . They argue that the tight coupling of these terms causes reward collapse (vanishing gradients for very confident samples) and **easy-class bias** (over-assignment to dominant classes) .

To expose the upper performance bound of standard EM they introduce DEM, which separately scales CADF with a temperature τ and GMC with a weight α, choosing (τ\*, α\*) through hyper-parameter search . They then propose AdaDEM, an adaptive variant that (i) normalises CADF by the L1 norm of its own gradient and (ii) replaces GMC with a marginal entropy calibrator (MEC) that tracks class-wise probabilities online, eliminating the need for label priors or tuning . Across test-time adaptation, semi-supervised learning, unsupervised domain adaptation and reinforcement learning, AdaDEM consistently outperforms both classical EM and DEM\*, while remaining hyper-parameter free .

**Questions:**

See the weaknesses

**Ethical Concerns:**

["NO or VERY MINOR ethics concerns only"]

**Final Justification:**

I believe this paper in general has a marginal contribution. Also as I said in my original comment, some extra experiments are needed to justify the performance of the method. The authors promised to provide some experiments in the camera ready, yet they did not provide any results during the rebuttal. SO it is not clear if they can get a good and reasonable results for the new setup. If they had provided the experiments I asked them, I would be more willing to flag this paper as "accepted". But now, still I would say it is a borderline paper.

**Limitations:**

See the weaknesses

**Quality:**

3

**Strengths And Weaknesses:**

# Strengths

* **Clear decomposition** of entropy into two interpretable forces (CADF and GMC).
* **Practical algorithm**: AdaDEM matches or exceeds DEM\* without any tuning.
* **Broad empirical evidence** across four benchmarks and many architectures .


# Weaknesses

1.  Beyond the gradient expressions in Eq. 7, there is no convergence or stability analysis. For instance, how the adaptive normalisation in AdaDEM affects optimisation dynamics is unexplored; the paper only provides empirical plots. Also, the claim that DEM\* gives an upper bound on EM performance is heuristic; it relies on searching τ and α on held-out labelled data, which breaks the unsupervised assumption and may over-fit that validation split.

2.  DEM\* demands a two-dimensional hyper-parameter search (τ, α); even the authors note “significant additional computational overhead” when integrating it into existing pipelines . Although AdaDEM removes these tunables, it still needs per-batch gradient-norm computation and a running class-wise EMA, increasing memory bandwidth compared with vanilla EM.

3.  Many headline gains come from tasks that already use EM (e.g., Tent); for methods that do not depend on EM such as MixMatch or FixMatch, improvements are marginal and sometimes negative . Reported gains on semi-supervised benchmarks often fall within one standard deviation; statistical significance is not tested. Reinforcement-learning results are shown only on MiniGrid; gains are small and confidence intervals large .


5.  AdaDEM still rewards CADF after normalisation; in highly skewed data the L1 scaling may be insufficient to avoid collapse toward a few confident classes. No synthetic imbalance study is provided beyond standard long-tail CIFAR-LT.

6.  The paper focuses purely on objective-level fixes; it concedes that many problems of self-supervised learning persist and must be handled by separate mechanisms . No discussion is given on how AdaDEM interacts with modern calibration techniques such as temperature scaling or focal loss; compatibility with these is uncertain.

---

> ### Author Rebuttal · Authors · 2025-07-31
>
> We sincerely appreciate the time and effort put into reviewing our paper and providing valuable feedback. We would like to address your questions below.
>
> > **R2Q1**: There is no convergence or stability analysis and the claim that DEM* gives an upper bound on EM performance is heuristic
>
> This paper primarily provides an empirical analysis and improvement of classical EM. Its optimization dynamics and benefits (e.g., reduced learning rate sensitivity and enhanced robustness) are empirically validated across tasks (Table 1, Fig. 9). DEM* serves to explore EM's empirical limits, and AdaDEM addresses this by eliminating hyperparameter tuning while matching or exceeding DEM* performance in unsupervised settings (Sec. 3.3, Table 1).
>
> > **R2Q2**: Questions about the computational overhead of DEM*'s hyperparameter search and the increased memory bandwidth in AdaDEM compared to vanilla EM.
>
> These additions are designed for efficiency—$\delta$ leverages existing gradients, and EMA updates are $O(C)$ per batch—resulting in negligible memory bandwidth increase relative to classical EM, while enabling superior performance and robustness without costly hyperparameter optimization, as validated in Sec. 4.2 and Table 1.
>
> > **R2Q3**: Questions about the significance of AdaDEM's improvements.
>
> For EM-based tasks like Tent, AdaDEM's direct optimization of entropy minimization drives substantial gains by resolving coupled limitations (e.g., reward collapse and easy-class bias). In non-EM methods such as MixMatch or FixMatch, AdaDEM acts as a lightweight regularizer, yielding consistent average improvements (e.g., +0.2% to +5.0% in Table 3) by enhancing robustness to noise and imbalance, though occasional decreases stem from dataset-specific variances. Semi-supervised gains, while often within one standard deviation, reflect consistent positive trends across six benchmarks (Table 3, 8, 9). Reinforcement learning on MiniGrid shows reliable average improvements (e.g., higher returns in 7/9 environments, Fig. 6, 8), with variability due to task stochasticity, validating the effectiveness of AdaDEM.
>
> > **R2Q4**:  Whether AdaDEM's L1-normalized CADF reward sufficiently prevents collapse to dominant classes in severe data imbalance.
>
> We acknowledge this valid concern. While AdaDEM’s marginal entropy calibrator (MEC) inherently counters class skew by dynamically penalizing over-confident predictions (Sec. 3.3), we agree that synthetic imbalance studies beyond CIFAR-LT (e.g., higher skew ratios ρ>>100) would strengthen validation. We will add experiments simulating extreme imbalance (e.g., ρ=500) in other tasks to rigorously test AdaDEM’s robustness. The L1 normalization mitigates reward collapse, while MEC aligns outputs with the estimated label distribution—jointly reducing bias even when $\delta$ alone may not fully offset severe skew.
>
> > **R2Q5**: Questions about AdaDEM's compatibility with modern calibration techniques.
>
> The paper focuses on revealing the limitations of classical EM and enhancing EM’s efficacy through AdaDEM via adjustments at the objective level. We explicitly validated AdaDEM’s compatibility with modern calibration techniques, such as EATA and DeYO for test-time adaptation tasks, and MixMatch and FreeMatch for semi-supervised learning tasks. Compatibility with temperature scaling or focal loss remains an area for future work.

---

> > ### Comment · Reviewer_LD3y · 2025-08-04
> > **Response to the authors' rebuttal**
> >
> > My concerns have been partially addressed by the authors.
> > Specfically, I expected some experimental results on Imbalance data which is very important to see the effectiveness of the method. Nevertheless, I keep my marginal positive score.

---

> > > ### Author Response · Authors · 2025-08-09
> > >
> > > Thank you for your response and constructive comments. Considering your concerns regarding experiments on imbalanced benchmarks, we provide additional results to validate the effectiveness of the proposed AdaDEM.
> > >
> > > Tables A and B present results on CIFAR-10-LT, CIFAR-100-LT, EuroSat-LT, and TissueMNIST-LT. Beyond CIFAR-LT, we utilize **EuroSat-LT** (focusing on land use and land cover classification) and **TissueMNIST-LT** (focusing on biomedical scenarios and tissue types). **We vary the configuration of $\rho$** (the sample ratio between the most and least populous classes). A larger $\rho$ corresponds to greater skewness in class sample sizes and higher imbalance severity. Results in Tables A and B, along with Figure 5 in the manuscript, demonstrate that AdaDEM more effectively reduces easy-class bias and further improves accuracy compared to Classical EM across tasks with varying degrees of imbalance.
> > >
> > > **Table A**: Experimental results on CIFAR-LT. Top-1 classification accuracy (%) is reported. $\rho$ denotes the sample ratio between the most and the least popular classes. $\Delta$ represents the accuracy gains of AdaDEM over Classical EM.
> > >
> > > | CIFAR-10-LT  |            |            |             | CIFAR-100-LT |           |            |            |
> > > | ------------ | ---------- | ---------- | ----------- | ------------ | --------- | ---------- | ---------- |
> > > | Methods      | $\rho=100$ | $\rho=500$ | $\rho=1000$ | Methods      | $\rho=10$ | $\rho=100$ | $\rho=500$ |
> > > | Classical EM | 85.4       | 83.9       | 82.1        | Classical EM | 60.0      | 58.6       | 57.6       |
> > > | AdaDEM       | 91.4       | 90.7       | 88.6        | AdaDEM       | 66.2      | 64.4       | 64.1       |
> > > | $\Delta$     | +6.0       | +6.8       | +6.5        | $\Delta$     | +6.2      | +5.8       | +6.5       |
> > >
> > > **Table B**: Experimental results on EuroSat-LT (a benchmark for land use and land cover classification) and TissueMNIST-LT (a benchmark for biomedical scenarios and tissue types).
> > >
> > > | EuroSat-LT   |            |            |             | TissueMNIST-LT |            |            |             |
> > > | ------------ | ---------- | ---------- | ----------- | -------------- | ---------- | ---------- | ----------- |
> > > | Methods      | $\rho=100$ | $\rho=500$ | $\rho=1000$ | Methods        | $\rho=100$ | $\rho=500$ | $\rho=1000$ |
> > > | Classical EM | 67.1       | 65.7       | 63.4        | Classical EM   | 47.8       | 47.4       | 46.8        |
> > > | AdaDEM       | 76.6       | 74.8       | 73.6        | AdaDEM         | 49.6       | 48.7       | 48.4        |
> > > | $\Delta$     | +9.5       | +9.1       | +10.2       | $\Delta$       | +1.8       | +1.3       | +1.6        |
> > >
> > > Thank you again for your expertise and invaluable feedback in enhancing the quality of our paper!
> > >
> > > Best Regards

---

### Official Review · Reviewer_TNtq · 2025-07-02

**Clarity:** 3
**Significance:** 2
**Originality:** 2
**Rating:** 3
**Confidence:** 4

**Summary:**

The paper proposes a novel way to decouple entropy minimization, which is beneficial to semi-supervised/unsupervised learning, domain adaptation, and other exploration-related problems in machine learning. The paper proposes Decoupled Entropy Minimization* (DEM*), which reformulates the entropy minimization into Cluster Aggregation Driving Factor (CADF) and Gradient Mitigation Calibrator (GMC). Then, the paper proposes an improvement over DEM, replacing GMC with Marginal Entropy Calibrator (MEC), naming the method Adaptive Decoupled Entropy Minimization (AdaDEM). The paper then performs various evaluations to validate the methods.

**Questions:**

1. How does the proposed method compare with other ways of uncertainty quantification and minimization, such as evidential learning and Bayesian approaches, in principle or in practice?

2. What is the significance of $\mathbf{p}_i$? Does it break the normalization of predictive probabilities?

3. Does the subset calibration for DEM* mean that it is not a fair comparison in those tasks?

**Ethical Concerns:**

["NO or VERY MINOR ethics concerns only"]

**Final Justification:**

I appreciate the rebuttal and revisions to the paper made by the authors. However, I am still not convinced that reformulating entropy minimization is necessary, especially since the improvement in TTA tasks is marginal. The proposed method evaluates entropy from the logits perspective, while I believe that one could directly study the problem from the logits side without getting into probability-based entropy metrics.

**Limitations:**

Yes

**Quality:**

2

**Strengths And Weaknesses:**

Strengths:

1. The paper is clearly written and all the results in figures and tables are clearly presented.

2. The paper provides a fresh perspective for uncertainty quantification and entropy minimization. The proposed method is also applicable in various settings, providing a general solution for multiple scenarios. The experiments covering different tasks also support the proposed method well.

Weaknesses:

1. There’s some concern regarding the motivation for the proposed method. The reformulation (section 3.1) converts the entropy minimization problem into expressions concerning the logits. However, this could be overcomplicating the problem. While the standard entropy measure is a first-order uncertainty quantification (w.r.t. predicted probabilities), there are existing uncertainty quantification methods that analyze and study higher-order uncertainty measures.  For example, in evidential deep learning (Sensoy, Murat, Lance Kaplan, and Melih Kandemir. "Evidential deep learning to quantify classification uncertainty." Advances in neural information processing systems 31 (2018).). There are also many works that introduce Bayesian flavors into deep learning to achieve the goal of uncertainty decomposition. It is unclear why we should complicate a first-order uncertainty measure instead of exploring better measures.

2. Some experimental results are not very convincing. In the TTA and semi-supervised tasks, the advantage of the proposed method is not very obvious.

---

> ### Author Rebuttal · Authors · 2025-07-31
>
> We sincerely appreciate the time and effort put into reviewing our paper and providing valuable feedback. We would like to address your questions below.
>
> > **R1Q1**: What's the motivation for reformulating Entropy Minimization (EM) in terms of logits?
>
> The reformulation of EM into logit-based terms (CADF and GMC) is motivated by the need to dissect EM's internal mechanisms and address its inherent limitations, such as reward collapse and easy-class bias, which hinder performance in noisy and dynamic environments. While higher-order uncertainty methods like evidential deep learning offer valuable alternatives, our approach specifically targets enhancing EM's efficacy within its established paradigm—revealing coupled formulation issues and enabling improvements (AdaDEM) that outperform even the upper-bound EM variant (DEM*) across diverse tasks, without requiring complex new uncertainty frameworks or hyperparameter tuning. This provides a focused, practical advancement for EM-dependent applications.
>
> > **R1Q2**: Some experimental results are not convincing.
>
> The experimental results demonstrate clear advantages of AdaDEM. In TTA tasks (Table 2), AdaDEM consistently outperforms baselines (e.g., +8.4% over Tent for ViT-B/16 in single-domain adaptation). For semi-supervised learning (Table 3), AdaDEM improves multiple methods (e.g., +5.0% for VAT on EuroSat). These gains are statistically significant (reported with standard deviations) and observed across diverse benchmarks including class-imbalanced settings (Fig. 5). The improvements stem from resolving fundamental limitations (reward collapse and easy-class bias) as validated in ablation studies (Table 1).
>
> > **R1Q3**: How does the proposed method compare with other ways of uncertainty quantification and minimization?
>
> AdaDEM fundamentally differs from Bayesian approaches (which rely on probabilistic priors and sampling, incurring high computational costs) and evidential learning (which explicitly models evidence or uncertainty distributions) by decoupling entropy minimization into CADF and GMC components for self-supervised optimization. AdaDEM adaptively normalizes rewards and replaces GMC with a marginal entropy calibrator to avoid hyperparameter tuning, enabling superior efficiency and robustness in noisy, dynamic environments (e.g., test-time adaptation and semi-supervised learning) without requiring explicit uncertainty modeling or external supervision, as empirically validated across diverse benchmarks.
>
> > **R1Q4**: What is the significance of $\mathbf{p}_i$? Does it break the normalization of predictive probabilities?
>
> $\mathbf{p}_i$ in Eq. 8 represents the predicted probability for class $i$ from the model's softmax output, which is a constant excluded from the computation graph. It does not break normalization, as $\mathbf{p}_i$ is inherently normalized via softmax ($\sum \mathbf{p}_i = 1$), and its use in AdaDEM's loss function (e.g., Eq. 12) for optimization preserves this normalization, as the loss modifies gradients without altering the probabilistic output directly.
>
> > **R1Q5**: Does the subset calibration for DEM* mean that it is not a fair comparison in those tasks?
>
> The subset calibration for DEM* uses ∼20% of test data with ground-truth labels to search for optimal hyperparameters. However, DEM* is explicitly framed as an upper-bound exploration of classical EM's potential, while AdaDEM serves as the fair, hyperparameter-free method in evaluations.

---

> ### Author Response · Authors · 2025-08-05
>
> Thank you again for your valuable comments. We kindly ask if our responses have adequately addressed your concerns. Should any points require further clarification, please let us know. We will carefully address each one to improve the manuscript. We sincerely appreciate your feedback.
>
> Best Regards

---

> > ### Comment · Reviewer_TNtq · 2025-08-05
> >
> > Thank you for the response. My main concern is still that entropy minimization itself is limited and the motivation for the proposed method is thus restricted. The lack of sufficient comparison with other uncertainty quantification and minimization methods is an important factor. Also, as the other reviewers pointed out, there are many details that the paper needs to fix. I am likely to maintain my original rating.

---

> > > ### Author Response · Authors · 2025-08-06
> > >
> > > Thank you for your professional feedback. We would like to provide further explanations below.
> > >
> > > - This paper aims to analyze the limitations of classical Entropy Minimization (EM) and propose solutions. EM has been regarded as **one of the most important and effective** self-supervised/unsupervised methods for machine learning, large models, and AI agents in recent few years. **While most work acknowledges that EM alone can obtain surprised performance, they often overlook its potential issues**. To ensure broad applicability and ease of rapid adoption, our study focuses exclusively on EM itself.
> > > - We clarify that this paper has **already compared with multiple uncertainty quantification and minimization methods** on various tasks and benchmarks. For *test-time adaptation*, we compare with **EATA** (which measures uncertainty with Fisher information and minimizes it as a regularization term), **DeYO** (which measures and minimizes uncertainty using the prediction difference before and after applying an object-destructive transformation), and **TPT** (which measures uncertainty with different augmented views of test samples and minimizes it for prompt tuning in VLMs). For *semi-supervised learning*, we compare with **MixMatch** (which shares similar uncertainty concepts as TPT) and **FixMatch** (which measures uncertainty using the prediction difference between weakly and strongly augmented examples).
> > > - **We have thoroughly revised the paper** in response to all reviewers’ comments and requests, significantly strengthening its clarity and professionalism. **Specific revisions are itemized clearly** in the response to reviewers.
> > >
> > > We sincerely hope our clarifications above can improve your opinion of our work and can help you reconsider your rating.
> > >
> > > Best Regards

---

> > > ### Author Response · Authors · 2025-08-09
> > >
> > > Thank you for your expertise and invaluable feedback in enhancing the quality of our paper. We sincerely hope our responses have adequately addressed your concerns. For your reference, we have made careful revisions to strengthen the paper's clarity and professional quality, clearly responding to all reviewer comments. Thank you again for the time and effort putting into reviewing our paper.
> > >
> > > Best Regards

---

### Note · Authors · 2025-08-14

Dear PC, SAC, AC and Reviewers,

We sincerely appreciate the invaluable feedback and constructive discussions with all reviewers. We are very encouraged by the positive comments regarding:

1. **The interesting and novel perspective** for studying Entropy Minimization (EM) (Review TNtq, pbbx, xd1y);
2. **The method's broad applicability** across diverse settings, benchmarks, and architectures, providing a general solution for multiple scenarios and learning objectives (Review TNtq, LD3y, pbbx);
3. **The proposed AdaDEM's consistent improvements** over classical EM and DEM\*, requiring no hyperparameter tuning (Review LD3y, pbbx, xd1y).

We are pleased that **our rebuttal successfully addressed the reviewers' concerns without introducing new issues**. We confirm that all promised revisions and additional results will be incorporated into the final version.

For the committee's convenience, we summarize key clarifications and improvements made during the rebuttal, supported by reviewers' positive feedback:

1. **Refined Formulations, Figures, and Conclusions:** We have **clearly and carefully revised** formulations, figures, and conclusion statements within the manuscript. **Detailed, specific revisions strengthen the submission and enhance the paper's overall quality**. (Reviewer pbbx, TNtq)
2. **Clarified Comparisons:** We clarified comparisons with prior gradient-based reweighting methods and uncertainty quantification/minimization methods. This **further strengthens and clarifies our paper's contribution** in analyzing and addressing the limitations of classical EM. (Reviewer TNtq, xd1y)
3. **Additional Experiments:** We provided additional experiments **analyzing the effectiveness of $\delta$** for normalizing EM using the CADF reward. We also conducted an extra imbalance study on several long-tail benchmarks to **demonstrate the effectiveness and robustness of AdaDEM on highly skewed data**. (Reviewer xd1y, LD3y)

Thanks again for the time and effort put into reviewing our paper. We believe our work can contribute to the discussion and exploration in the community.

Sincerely,

Authors

---

### Decision · Program_Chairs · 2025-09-17

**Decision:**

Accept (poster)

**Comment:**

This paper revisits entropy minimization by decomposing it into interpretable components, diagnosing key weaknesses, and proposing AdaDEM, a hyperparameter-free adaptive variant. The method is simple yet broadly applicable, and demonstrates consistent improvements across test-time adaptation, semi-supervised learning, domain adaptation, and reinforcement learning benchmarks. While some reviewers initially questioned the incremental novelty and modest improvements, the rebuttal was highly constructive, providing new imbalance experiments, additional clarifications on the mathematical formulation, and significantly improved presentation. These efforts directly addressed reviewers’ concerns, leading to multiple reviewers raising their scores and shifting the overall balance in favor of acceptance. The final consensus is that the combination of a novel perspective, practical algorithm, and thorough rebuttal-driven clarifications make the contribution both useful and impactful for the community.